# Meta-Learning the Search Distribution of Black-Box Random Search Based Adversarial Attacks

**Maksym Yatsura**
Bosch Center for Artificial Intelligence
University of Tübingen
`maksym.yatsura@de.bosch.com`

**Jan Hendrik Metzen**
Bosch Center for Artificial Intelligence
`janhendrik.metzen@de.bosch.com`

**Matthias Hein**
University of Tübingen
`matthias.hein@uni-tuebingen.de`

## Abstract

Adversarial attacks based on randomized search schemes have obtained state-of-the-art results in black-box robustness evaluation recently. However, as we demonstrate in this work, their efficiency in different query budget regimes depends on manual design and heuristic tuning of the underlying proposal distributions. We study how this issue can be addressed by adapting the proposal distribution online based on the information obtained during the attack. We consider Square Attack, which is a state-of-the-art score-based black-box attack, and demonstrate how its performance can be improved by a learned controller that adjusts the parameters of the proposal distribution online during the attack. We train the controller using gradient-based end-to-end training on a CIFAR10 model with white box access. We demonstrate that plugging the learned controller into the attack consistently improves its black-box robustness estimate in different query regimes by up to 20% for a wide range of different models with black-box access. We further show that the learned adaptation principle transfers well to the other data distributions such as CIFAR100 or ImageNet and to the targeted attack setting[1].

## 1 Introduction

It was demonstrated that despite their impressive performance in various tasks, neural networks are susceptible to small imperceptible perturbations in the input called *adversarial examples* [4]. This is a concerning issue for real-world deployment of deep learning approaches, especially in safety-critical domains such as autonomous driving [5]. But apart from practical concerns, adversarial examples serve as a tool to better understand the true nature of artificial neural networks and capture their inherent properties and differences from their biological counterparts [6].

Evaluating adversarial robustness is usually formulated as a constrained optimization problem [4]. However, finding the exact solution is typically intractable [7, 8]. Therefore, in practice one often resorts to approximate methods called *adversarial attacks* that try to find adversarial examples within a small number of iterations. A number of different adversarial attacks were proposed [8–10] that can be distinguished into white- and black-box attacks. In white-box attacks, one assumes full access to the model architecture and weights [8]. However, as white-box attacks typically rely on the gradient, gradient obfuscation, which does not eliminate the existence of adversarial examples but makes it significantly harder to find them gradient-based, can be a severe obstacle for certain attacks [11].

---

[1]The code is available at `https://github.com/boschresearch/meta-rs`

35th Conference on Neural Information Processing Systems (NeurIPS 2021).

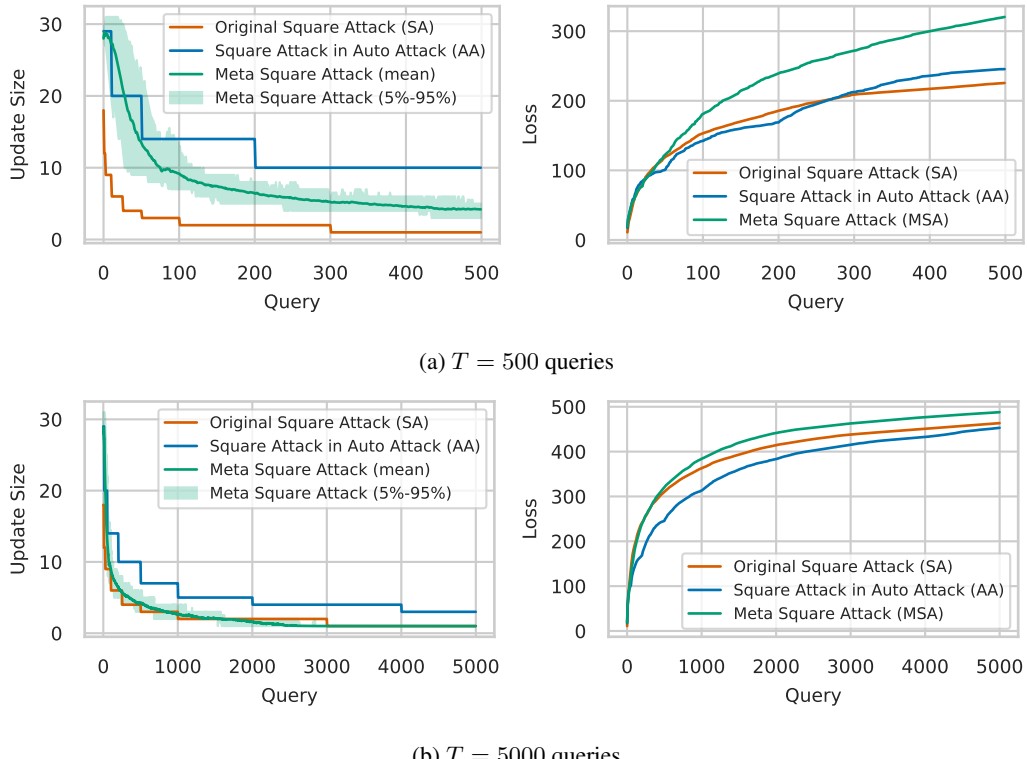

(a) $T = 500$ queries

(b) $T = 5000$ queries

Figure 1: (Left) The schedules for SA [1] (which scales accordingly for different query budgets) and AA [2] compared to Meta Square Attack (MSA) proposed in this work. MSA adapts the update size during the course of the attack for each image. We illustrate the mean and the percentile range for a given set of attacked images. (Right) The maximization of the loss for the model of Ding et al. [3] on 100 CIFAR10 images. MSA outperforms SA and AA significantly in terms of the achieved loss.

Therefore, black-box attacks that use only limited information from the model such as the class scores or the final decision often provide an additional perspective on the true robustness of the model [12–21] and are thus recommended for reliable robustness evaluation [2, 22].

A very promising direction in the field of black-box adversarial attacks are randomized search schemes for crafting adversarial examples [1, 23, 24]. Combining random search with specific update proposal distributions allows to achieve state-of-the-art black-box efficiency for different threat models such as $\ell_\infty$ and $\ell_2$ [1], $\ell_1$ [25], $\ell_0$, adversarial patches, and adversarial frames [24]. Despite the conceptual simplicity of these methods, the main disadvantage of random search based methods is that the construction of a suitable proposal distribution requires significant manual design and is crucial for competitive performance (see also Figure 1).

In this work, we propose a method that allows to circumvent the fine-tuning and reduce the amount of manual design in random search based attacks. We use gradient-based meta-learning to automatically optimize controllers for schedules and proposal distribution on models with white-box access. After meta-training, the controllers can be plugged into a random search attack substituting manually designed schedules and proposal distributions. Importantly, once meta-trained, the controllers do not require any gradient access and can thus be used in a fully black-box setting and without being affected by gradient obfuscation. We consider the proposed methodology for the case of Square Attack for the $\ell_\infty$ and $\ell_2$ threat models [1]. We meta-train controllers for update size and color on an adversarially trained [26] ResNet18 [27] model with white-box access and apply them to many different models from the RobustBench model zoo [28] that we treat as black-boxes. Since query efficiency is of crucial importance in black-box adversarial attacks, we also study the method for different query regimes ranging from several hundreds to several thousands. Depending on the query regime and the attacked model we obtain up to 20% improvement with respect to the baseline schedules proposed by Andriushchenko et al. [1] and Croce and Hein [2].

In short, we make the following contributions:

- We frame adversarial attack optimization as a meta-learning problem (Section 3.2).
- We formalize the gradient-based meta-learning for the state-of-the-art Square Attack and propose Meta Square Attack (Section 3.3).
- We meta-train Meta Square Attack (MSA) on a CIFAR10 [48] model with white-box access and show that MSA improves robust accuracy by up to 5.6% on a vast range of CIFAR10 models with black-box access with respect to the hand-designed search distributions proposed in previous work [1, 2] for the $\ell_\infty$ and $\ell_2$ threat models (Section 4).
- We show that Meta Square Attack generalizes well to different datasets and to the targeted attack setting. It achieves up to 20% better robust accuracy compared to the state-of-the art baseline [1] for attacking models on CIFAR100 and ImageNet (Section 4.2).

## 2 Related Work

**Black-box adversarial attacks:** Existing approaches to black-box robustness evaluation can be divided into several groups based on their mode of interaction with the target model. Transfer-based attacks [17, 29] rely on having a model with white-box access that is similar to the targeted model. If this is the case, one can generate adversarial perturbations for this substitute model and transfer them in a black-box setting. A downside of such approaches is that similarity between the models is a crucial factor. Without it the efficiency of transfer attacks is drastically reduced [29]. Decision-based attacks [13, 20, 30] applied to classifiers assume that one can submit queries to the model and receive the predicted class label. Score-based attacks [1, 12–19] consider broader access to the model and assume the attack receives predicted scores for all the classes. Prior work [15, 31, 32] use this information to estimate a gradient and apply gradient descent. Alzantot et al. [19] propose a derivative-free approach based on genetic algorithms. Several works use the empirical observation that successful $\ell_\infty$ perturbations are located at the corners of the color cube, therefore instead of continuous optimization one can reduce the problem to a discrete one [1, 12, 24, 33].

**Meta-learning and adversarial robustness:** The closely related fields of meta-learning [34] and learning to optimize [35] have been employed in the field of adversarial robustness. The idea of learned optimizers [36, 37] was applied to finding adversarial examples in white-box [38] and black-box [39] settings. Meta-learning [40, 41] was also used to improve zeroth-order gradient estimation and allow better query efficiency of black-box attacks [42]. Besides that there is the recent work [43] on automating the existing AutoAttack [2] framework for robustness evaluation. Meta-learning has also been used as part of adversarial training [38, 44] for increasing adversarial robustness.

**Square Attack:** Square Attack (SA) was proposed by Andriushchenko et al. [1] and combines classical random search with a heuristic design of the update rule. This design depends on the geometry of the perturbation set and therefore differs for $\ell_\infty$- and $\ell_2$-attacks and was later extended also to $\ell_1$ [25]. As the adversarial attack problem is highly non-convex, a good initialization can significantly improve query efficiency. SA uses a stripe initialization motivated by empirical findings [45]. The design principle of the attack is based on the observation that the strongest perturbations are usually found on the boundary of the feasible set [18]. For the $\ell_\infty$-case, updates are sampled as squares parametrized by square size, square color, and position of the square. Performance of SA depends heavily on how color, position, and square size are chosen, which requires manual design.

The square size schedules employed by previous work are relatively sophisticated (indicating non-trivial manual design): Andriushchenko et al. [1] proposed a schedule parametrized by $p^0 \in [0, 1]$ (the fraction of image pixels to be modified by a square in the first query) and the total query budget $T$, where $p^t$ is halved at $\{0.1, 0.5, 2, 10, 20, 40, 60, 80\}\%$ of the total query budget. For evaluation on CIFAR10, Andriushchenko et al. [1] suggest different values of $p^0$ and report $p^0 = 0.3$ as a default choice. Croce and Hein [2] proposed a different schedule for SA to be used in AutoAttack: they use $p^0 = 0.8$ and $T = 5000$ but fix the halving points of $p^t$ as if $T = 10000$. An illustration of the two schedules for $T = 500$ and $T = 5000$ can be seen in Figure 1.

The second aspect that characterizes SA is the distribution from which position and color of the next square are sampled. For positions, the distribution is uniform over all positions for which a square of a given size would be fully contained in the input. For colors, SA always generates points on the boundary of the perturbation set (corners of the color cube) and uses a uniform distribution over the $2^c$ different colors (with $c$ being number of channels, typically $c = 3$).

Thus SA either relies on extensive manual design (for the square size schedules) or resorts to simple baseline choices (such as the uniform distributions over colors and positions) which might be suboptimal. In this work, we show that meta-learning SA consistently improves the already strong performance of SA (see Section 4.2) with very little manual design and identify non-trivial patterns that increase attack efficiency (see Section 4.3).

## 3 Method

In the following, we briefly introduce the formulation of the optimization problem for an adversarial attack, rephrase it in the from of a meta-learning problem, and introduce our method for learning the search distribution of a specific random search based black-box adversarial attack, namely Square Attack [1]. We denote this method as Meta Square Attack (MSA).

### 3.1 Adversarial Robustness Evaluation

Let $K$ be the number of classes in a classification problem and $f : [0,1]^d \to \Delta^{K-1}$ be a classifier which maps a $d$-dimensional input $x$ to

$$\Delta^{K-1} = \left\{ (p_0, ..., p_{K-1}) \in \mathbb{R}^K \Big| \sum_{i=0}^{K-1} p_i = 1, \text{ and } p_i \geq 0 \text{ for } i = 0, ..., K-1 \right\},$$

which denotes the set of probability distributions over the $K$ discrete possible outcomes. For a label $y \in \{0, ..., K-1\}$, a loss function $l : \Delta^{K-1} \times \{0, ..., K-1\} \to \mathbb{R}$, a perturbation set $S$, and operator $a$, we define the robustness evaluation problem as:

$$V(f, x, y) = \max_{\xi \in S} l\big(f(a(x, \xi)), y\big) \tag{1}$$

For $S = \{\xi \mid ||\xi||_p \leq \epsilon\}$ and $a(x, \xi) = \Pi_{[0,1]^d}(x + \xi)$ (where $\Pi_{[0,1]^d}$ is the projection onto $[0,1]^d$), one obtains the standard $\ell_p$ ball threat model for images. Assuming that the loss function $l$ and operator $a$ are fixed, we denote $L(f, x, y, \xi) := l(f(a(x, \xi)), y)$ as a functional that one needs to maximize in robustness evaluation.

Since exact maximization of Equation (1) is intractable in the general case [7], we consider a (potentially non-deterministic) procedure $\mathcal{A}_\omega(L, f, x, y)$ called adversarial attack. This attack $\mathcal{A}_\omega$ is parametrized by hyperparameters $\omega$ and designed with the intention that $\xi^\omega \sim \mathcal{A}_\omega(L, f, x, y)$ with $\xi^\omega \in S$ becomes an approximate solution (tight lower bound) of $V(f, x, y)$, that is $V(f, x, y) - L(f, x, y, \xi^\omega)$ should become small (in expectation). Optimizing the hyperparameters of the attack via $\max_\omega \mathbb{E}_{\xi^\omega \sim \mathcal{A}_\omega(L, f, x, y)} L(f, x, y, \xi^\omega)$ can allow a tighter lower bound of $V(f, x, y)$. Unfortunately, this maximization is still intractable typically, for instance when $f$ is a black-box (no gradient information is available), the number of queries to $f$ per data $(x, y)$ is limited, or $\mathcal{A}_\omega$ has high variance.

### 3.2 Black-box Adversarial Attack Optimization as a Meta-learning Problem

We now frame optimization of the adversarial attack in the query-restricted black-box setting as a *meta-learning* problem. We follow the taxonomy proposed in the recent survey on meta-learning by Hospedales et al. [34].

First, we formulate our **meta-objective** (the specification of the goal of meta-learning): we assume data to be governed by a distribution $(x, y) \sim \mathcal{D}$ and classifiers defined on this data, which need to be evaluated, by a distribution $f \sim \mathcal{F}$. Our meta-objective is to find parameters $\omega^*$ of the attack $\mathcal{A}_\omega$ that maximize the lower bound $L(f, x, y, \xi^\omega)$ of $V(f, x, y)$ in expectation across models $f \sim \mathcal{F}$, data $(x, y) \sim \mathcal{D}$, and the stochastic attack $\xi^\omega \sim \mathcal{A}_\omega$:

$$\omega^* = \underset{\omega}{\operatorname{argmax}} \, \mathbb{E}_{f \sim \mathcal{F}} \, \mathbb{E}_{(x,y) \sim \mathcal{D}} \, \mathbb{E}_{\xi^\omega \sim \mathcal{A}_\omega(L, f, x, y)} L(f, x, y, \xi^\omega) \tag{2}$$

Here, the expensive optimization of $\omega^*$ is amortized across models and data. More specifically, we assume finite sets of data $D = \{(x_i, y_i)_{i=1}^N \mid (x_i, y_i) \sim \mathcal{D}\}$ and classifiers $F = \{f_j \mid f_j \sim \mathcal{F}\}$

are available. Moreover, we assume the $f_j \in \mathcal{F}$ allow white-box access and quasi-unrestricted number of queries. The sets $D$ and $F$ can be used during *meta-training* of the attack, that is the objective during meta-training becomes $\omega^* = \arg\max_\omega R(F, D, \omega)$ with $R(F, D, \omega) = \sum_{f \in F} \sum_{(x,y) \in D} L(f, x, y, \xi^\omega)$ and $\xi^\omega \sim \mathcal{A}_\omega(L, f, x, y)$. However, the ultimate goal of meta-learning is to apply $\mathcal{A}_{\omega^*}$ during *meta-testing* to unseen $(x, y) \sim \mathcal{D}$ and unseen $f \sim \mathcal{F}$ that allow only black-box and query-limited access, and maximize $\mathbb{E}_{\xi^\omega \sim \mathcal{A}_\omega(L, f, x, y)} L(f, x, y, \xi^\omega)$, that is: the attack needs to generalize across models and data.

Next, we define the **meta-representation**, that is how the adversarial attack $\mathcal{A}_\omega$ is designed and parametrized such that generalization across $f \sim \mathcal{F}$ is effective. In this work we focus on random search based adversarial attacks for black-box robustness evaluation since they have achieved strong results in prior work and are amenable to meta-learning. Let $\mathcal{A}_\omega$ be a random search based attack with a query budget limited by $T$. Then an adversarial perturbation $\xi^\omega = \xi^T \sim \mathcal{A}_\omega(L, f, x, y)$ is obtained using the following iterative procedure:

$$\xi^0 \sim \mathcal{D}^0; \quad \xi^{t+1} = \underset{\xi \in \{\xi^t, \Pi_\mathcal{S}(\xi^t + \delta^{t+1})\}}{\arg\max} L(f, x, y, \xi); \quad \delta^{t+1} \sim \mathcal{D}_\omega(t, \xi^0, \delta^0, \dots, \xi^t, \delta^t), \quad (3)$$

where $\Pi_\mathcal{S}$ corresponds to the projection onto the perturbation set $\mathcal{S}$.

That is, we assume a fixed distribution $\mathcal{D}^0$ for initializing the perturbation $\xi^0$ but a meta-learnable $\mathcal{D}_\omega$ for the update proposals $\delta^{t+1}$. Importantly, $\mathcal{D}_\omega$ depends on the entire attack trajectory up to step $t$. Since this trajectory contains implicitly information on the classifier $f$ when applied to data $(x, y)$, our meta-learned random search attack can adapt to the classifier and data at hand. We provide more details on $\mathcal{D}_\omega$ for the specific case of Square attack [1] in Section 3.3.

The **meta-optimizer** (how we optimize the meta-objective) in our case assumes that both the loss function $l$ and $\mathcal{A}_\omega$ are (stochastic) differentiable with respect to $\omega$ or we can find differentiable relaxations (as we will discuss in Section 3.3). Thus the meta-parameters $\omega$ can be optimized using stochastic gradient descent on mini-batches $B \subseteq D$ based on the (stochastic) gradient

$$g = \nabla_\omega R(F, D, \omega) = \sum_{f_j \in F} \sum_{(x_i, y_i) \in B \subseteq D} \nabla_\omega L(f_j, x_i, y_i, \xi_{i,j}), \quad (4)$$

where $\xi_{i,j} \sim \mathcal{A}_\omega(L, f_j, x_i, y_i)$. However, due to the stochasticity of $\mathcal{A}_\omega$ induced by $\mathcal{D}^0$ and $\mathcal{D}_\omega$, the discrete $\arg\min$ in the update step of $\mathcal{A}_\omega$, and the length $T$ of the unrolled optimization (often in the order of hundreds to thousands queries), $g$ would have very high variance and typically one would also face issues with vanishing or exploding gradients. To address this, we propose using a greedy alternative instead:

$$g = \frac{1}{T} \sum_{f_i} \sum_{(x_j, y_j)} \sum_{t=1}^{T-1} \nabla_\omega L(f_i, x_j, y_j, \Pi_\mathcal{S}(\xi^t + \delta^{t+1})). \quad (5)$$

Importantly, even though $\xi^t$ depends on $\omega$ for $t > 0$, we do not propagate gradients with respect to $\omega$ through $\xi^t$, that is we set $\nabla_\omega \xi^t := 0$. By this, the gradient corresponds to optimizing $\mathcal{D}_\omega$ in a myopic way, such that proposals $\delta^{t+1} \sim \mathcal{D}_\omega(t, \xi^0, \delta^0, \dots, \xi^t, \delta^t)$ are trained to maximally increase the immediate loss in step $t+1$. While this introduces a bias of acting myopic and greedy, it works reasonably well in practice.

We can now rewrite $\nabla_\omega L(f_i, x_j, y_j, \xi^t + \delta^{t+1}) = \nabla_{\delta^{t+1}} L(f_i, x_j, y_j, \xi^t + \delta^{t+1}) \nabla_\omega \delta^{t+1}$. We note that $\nabla_\omega \delta^{t+1}$ can be a very high variance estimate of $\nabla_\omega \mathcal{D}_\omega(t, \xi^0, \delta^0, \dots, \xi^t, \delta^t)$ depending on the stochasticity of $\mathcal{D}_\omega$. However, as it is an unbiased estimate and we average over very many steps $T$ and models and data, $g$ is still a sufficiently good estimate in practice.

### 3.3 Meta Square Attack

In this section, we demonstrate how the proposed meta-learning approach can be applied to Square Attack (SA) [1] with $\ell_\infty$ threat model. We denote the resulting meta-learned attack as *Meta Square Attack* (MSA). We keep $\mathcal{D}^0$ as the stripe initialization from SA and focus on meta-learning $\mathcal{D}_\omega$ as it governs all but the first step. As discussed in Section 2, sampling $\delta^{t+1} \sim \mathcal{D}(t)$ in SA proceeds by computing a square size (its width in pixels) $s_t = \pi^s(t) \in \{1, \dots, s_{max}\}$ and sampling a

position $(p_x, p_y) \sim \pi^p(s) \in \{1, \ldots, s_{max} - s\}^2$ and a color $c \sim \pi^c \in \{c_1, \ldots c_m\}$. In SA, $\pi^s$ is a heuristic schedule that depends on $t$ (and differs in prior work [1, 2]) and both $\pi^p$ and $\pi^c$ are uniform distributions. $\delta^{t+1}$ is then chosen to be zero everywhere except for a square of size $s$ at position $p$ with color $c$. Possible colors $c_i$ correspond to the eight corners of the RGB hypercube with $\ell_\infty$ norm $\epsilon$. That is $c_i \in (\pm\epsilon, \pm\epsilon, \pm\epsilon)$, while $s_{max}$ is chosen maximally with the constraint that all sampled squares must not exceed the image dimensions. We keep $\pi^p$ as uniform distribution, but meta-learn the controllers $\pi^s_{\omega_s}$ and $\pi^c_{\omega_c}$ with parameters $\omega = (\omega_s, \omega_c)$.

**Update size controller** $\pi^s_{\omega_s}$. We design $\pi^s_{\omega_s}$ as a multi-layer perceptron (MLP) with parameters $\omega_s$. The MLP outputs a scalar value $s' \in \mathbb{R}$ and we map this value to the actual update size via $s = \sigma(s') \cdot (s_{max} - 1) + 1$ with $\sigma(x) = 1/(1 + e^{-x})$. During meta-testing, we round the continuous $s \in [1, s_{max}]$ to a discrete value $\lfloor s \rfloor \in \{1, \ldots, s_{max}\}$. As this would block gradient flow during meta-training, we relax the square sampling in SA such that it supports continuous update sizes. The details of this relaxed sampling can be found in the Appendix A.3. Importantly, it is only conducted during meta-training and not in the final evaluation during meta-testing.

We provide two scalar inputs to the MLP $\pi^s_{\omega_s}$: (a) the current query $t$ encoded as $\log_2(\frac{t}{T} + 1)$ where $T$ is the maximal number of queries. It ensures that the input stays in the range $[0, 1]$ for $t \leq T$. We use $T = 5000$. (b) Let $r^{t+1} = H\left(L(f, x, y, \Pi_\mathcal{S}(\xi^t + \delta^{t+1})) - L(f, x, y, \xi^t)\right) \in \{0, 1\}$ be an indicator of whether adding $\delta^{t+1}$ at time $t+1$ improved the loss (for $H$ being the Heaviside step-function). The MLP gets as second input at time $t$ the value $R^t = \gamma R^{t-1} + (1 - \gamma)r^t/r^0$ with $R^0 = 1$. Here $\gamma$ is a decay term that controls how quickly past experience is "forgotten" and $r^0 = 0.25$ is a constant whose purpose is to ensure that the MLP's second input has a similar scale as the first (namely in $[0, 1]$). Intuitively, (a) allows to schedule update sizes based on time step of the attack (this information is also used in SA itself) while (b) allows adapting the update size based on the recent success frequency $R^t$ of proposals (for instance reducing the update size if few proposals were successful recently). Thus (b) allows the schedule to adapt to classifier $f$ and data $(x, y)$ at hand.

**Color controller** $\pi^c_{\omega_c}$. We design the color controller $\pi^c_{\omega_c}$ as a categorical distribution $c^t \sim \text{Cat}(\alpha_1^t, \ldots, \alpha_m^t)$, where each $\alpha_i^t \in \mathbb{R}$ is predicted by an MLP with weights $\omega_c$. The $m$ MLPs for the $\alpha_i^t$ share weights $\omega_c$ but differ in their inputs. In order to differentiate through the categorical distribution at meta-train time, we reparametrize the categorical distribution with Gumbel-softmax and draw discrete (hard) samples in the forward pass but treat them as soft samples in the backward pass [46, 47]. Additionally, we ensure that every color $c_i$ is sampled at least with probability $p^c_{min}$ by assigning $P(c_i) = p^c_{min} + (1 - mp^c_{min})P_{\text{Cat}(\alpha_1^t, \ldots, \alpha_m^t)}(c_i)$. This ensures continuous exploration of all $m$ colors.

The $m$ MLPs get two inputs: (a) the current query $t$ encoded as $\log_2(\frac{t}{T} + 1)$ (same encoding as for the step size controller and also same for all $m$ MLPs). (b) Information regarding the recent success frequency of proposals $\delta^t$ based on squares of the respective colors ($c^t = c_i$):
$R_i^t = \begin{cases} \gamma R_i^{t-1} + (1 - \gamma)r^t/r^0 & \text{if } c^t = c_i \\ R_i^{t-1} & \text{otherwise} \end{cases}$. This second input allows the controller to learn, e.g., to sample those colors more often that resulted in higher success frequency recently.

## 4 Experiments

We perform an empirical evaluation of Meta Square Attack (MSA). First, we consider the data distribution $\mathcal{D}$ of CIFAR10 [48] images and a classifier distribution $\mathcal{F}$ consisting of the classifiers robust with respect to the $\ell_\infty$-threat model. We use this setting for the meta-training as discussed in Section 4.1. We further consider how the controllers trained for these distributions generalize to working with the other data distributions of CIFAR100 and ImageNet and corresponding distributions of classifiers defined on this data. We also discuss the meta-training for the distribution of the classifiers robust with respect to the $\ell_2$-threat model in the Section A.5.

We compare the performance of MSA in 4 different query budget regimes to manually designed schedules for Square Attack proposed by Andriushchenko et al. [1] (denoted by SA) and Croce and Hein [2] (denoted by AA). The reason why we have chosen this evaluation mode instead of reporting accuracy and average number of queries for some single fixed budget is that the original SA approach proposes to scale the schedule to a given budget (Figure 1). Hence, it uses the knowledge of the attack budget and the schedule used for 500 queries is not a truncated version of the schedule used for

Table 1: MSA compared to SA [1] and AA [2] in the $\ell_\infty$-threat model with $\epsilon = 8/255$ on 1000 CIFAR10 test images. We report mean and standard error of robust accuracy across 5 runs with different random seeds. The full results for 16 CIFAR10 models can be found in Table 8.

| Model | Accuracy (%) | | Attack | Query budget | | | |
| | Clean | Robust | | 500 | 1000 | 2500 | 5000 |
|---|---|---|---|---|---|---|---|
| Wong et al. [51] | 83.34 | 43.21 | SA | 69.7±0.15 | 63.5±0.10 | 55.1±0.04 | 50.8±0.08 |
| | | | AA | 69.5±0.21 | 63.9±0.10 | 57.4±0.07 | 53.6±0.06 |
| | | | MSA | **63.9±0.12** | **59.1±0.09** | **53.0±0.16** | **49.8±0.08** |
| Huang et al. [52] | 83.48 | 53.34 | SA | 72.3±0.10 | 66.6±0.16 | 60.5±0.09 | 57.3±0.08 |
| | | | AA | 70.6±0.10 | 66.5±0.07 | 61.2±0.10 | 58.5±0.11 |
| | | | MSA | **66.1±0.10** | **62.9±0.15** | **58.7±0.05** | **56.8±0.08** |

5000 queries as it is done in AA. Since adapting the schedule to different query budgets is a crucial factor for the manually designed schedules that we consider as baselines, we choose the evaluation regime that allows to take this factor into consideration. Section 4.2 summarizes the experimental results (for more details see the Section A.1 in the Appendix) and Section 4.3 analyzes the behavior learned by the controllers.

## 4.1 Meta-Training and Controller Design

We meta-train the controller on a single robust model [2] $f \sim \mathcal{F}$ with white-box access (the "source model"). The source model was designed such that attackers could easily and cheaply acquire it themselves: the model has ResNet18 [27] architecture and was trained on the CIFAR10 training set using adversarial training [26] with the advertorch [49] package. Adversarial training was done using $\ell_\infty$-PGD attack with $\epsilon = 8/255$, fixed step size of 0.01, and 20 steps.

For both update size and color controllers, we use MLP architectures with 2 hidden layers, 10 neurons each, and ReLU activations. We purposefully did not finetune the MLP architecture. Meta-training was run on a set $D$ consisting of 1000 images from CIFAR10 test set (different from the ones used in evaluation of controllers in the next subsection) and Square Attack with a query budget of 1000 iterations. Therefore, controller behaviour on query regimes higher than 1000 are obtained by extrapolation of the behaviour learned for 1000 iterations. Both controllers were trained simultaneously for 10 epochs using Adam optimizer with batch size 100 and cosine step size schedule [50] with learning rate 0.03. The total loss improvement over the attack was used as meta-loss that we optimized in meta-training. We always run the attack on all images for the full budget $T$, since removing images from the attacked batch would cause discontinuity in the meta-loss. All computations including meta-training and evaluation of the controllers were performed on a single Nvidia Tesla V100-32GB GPU.

## 4.2 Evaluation

In this section, we evaluate how the Meta Square Attack (MSA) obtained by training on a CIFAR10 model with white-box access discussed in the Section 4.1 performs for different models and datasets.

Table 1 illustrates that MSA transfers well to two selected robust CIFAR10 models. Moreover, Table 2 reports results aggregated over a broad range of 16 robust CIFAR10 models from RobustBench (see Table 8 in the Appendix A.4 for details). We report mean, minimal, and maximal improvement across all the 16 models. We observe a consistent improvement for each considered query budget regime, which is especially pronounced for lower query regimes of 500 and 1000 queries. The improvements generalize to a budget of 5000 queries, which is five times higher than the one used during meta-training.

As described in Section 3.3, the input to $\text{MSA}_s$ and $\text{MSA}_c$ is not specific to the data distribution. Therefore, we also show that the adaptation principles learned on a CIFAR10 model transfer well to attacking models that not only have different architecture but also operate on significantly different

---

[2]Meta-training on more than one source model could improve generalization across models, but we found that even meta-training on a single model generalizes sufficiently well.

Table 2: Improvement in $l_\infty$-robust accuracy of our MSA with respect to the *best* of the previous Square Attack configurations, SA [1] and AA [2], in the setting of Table 1. The results are accumulated across 16 robust CIFAR10 models from RobustBench [28] (see Table 8 for full results).

| Query budget | 500 | | | 1000 | | | 2500 | | | 5000 | | |
|---|---|---|---|---|---|---|---|---|---|---|---|---|
| Improvement in | mean | min | max | mean | min | max | mean | min | max | mean | min | max |
| robust accuracy (%) | 4.29 | 3.1 | 5.6 | 3.87 | 2.7 | 5.4 | 1.63 | 0.9 | 2.1 | 0.38 | -0.1 | 1.0 |

Table 3: **Transfer to CIFAR100:** MSA trained on a CIFAR10 model consistently outperforms SA [1] and AA [2] on CIFAR100 (1000 images) in robust accuracy for the $\ell_\infty$-threat model with $\epsilon = 8/255$. Averaged across 3 runs with different random seeds.

| Model | Accuracy (%) | | Attack | Query budget | | | |
|---|---|---|---|---|---|---|---|
| | Clean | Robust | | 500 | 1000 | 2500 | 5000 |
| Wu et al. [53] | 60.38 | 28.86 | SA | 43.3±0.17 | 38.7±0.09 | 33.9±0.28 | 32.3±0.20 |
| | | | AA | 41.3±0.20 | 37.6±0.00 | 35.0±0.06 | 32.7±0.35 |
| | | | MSA | **37.8±0.07** | **35.5±0.12** | **33.1±0.03** | **32.2±0.03** |
| Cui et al. [54] | 70.25 | 27.16 | SA | 48.9±0.03 | 42.3±0.20 | 33.6±0.09 | 30.5±0.06 |
| | | | AA | 47.5±0.17 | 42.8±0.09 | 35.9±0.13 | 32.5±0.15 |
| | | | MSA | **42.6±0.13** | **37.8±0.12** | **32.5±0.30** | **30.1±0.09** |

data distributions: Table 3 demonstrates generalization of the learned controllers for attacking robust models for CIFAR100 [48]. We also consider the transfer to ImageNet [57] dataset that has significantly higher input dimension and number of classes than CIFAR10. In Table 4, one can see that MSA significantly improves the results even in the high extrapolation regime of 5000 queries. We also observe considerable improvement of the robust accuracy estimate for the targeted attacks on undefended ImageNet models (Table 5). Results for applying the size controller trained specifically for the $\ell_2$ threat model (together with the color controller trained for $l_\infty$) are in Table 6. We observe consistent improvement of about 3% robust accuracy for all of the considered query budgets. The magnitude of improvement for different datasets and threat models depend on two factors: how well the adaptation mechanism learned by our controllers generalizes in the given setting and how suitable the hand-designed search distributions of the baselines [1, 2] are for each particular problem. The consistency of the improvement indicates that the adaptive search distribution makes Meta Square Attack more efficient in the majority of the settings.

## 4.3 Analysis of Learned Controllers

As the learned controllers are black-boxes (implemented by MLPs), it may be non-trivial to understand their realized strategy. We present some analysis of the controllers' internal strategy based on their empirical behavior. Further analysis can be found in the Section A.4.

Table 4: **Transfer to ImageNet:** MSA trained on a CIFAR10 model consistently outperforms SA [1] and AA [2] on 1000 ImageNet validation set images in $l_\infty$-robust accuracy for $\epsilon = 4/255$. For SA we use $p^0 = 0.05$ as in Andriushchenko et al. [1]. Averaged across 3 different runs.

| Model | Accuracy (%) | | Attack | Query budget | | | |
|---|---|---|---|---|---|---|---|
| | Clean | Robust | | 500 | 1000 | 2500 | 5000 |
| resnet18 Salman et al. [55] | 52.5 | 25.0 | SA | 50.6±1.43 | 48.1±1.18 | 43.9±1.00 | 40.3±1.21 |
| | | | AA | 45.2±1.09 | 43.5±0.86 | 41.0±1.07 | 39.0±1.21 |
| | | | MSA | **43.3±1.00** | **41.7±0.94** | **39.1±1.23** | **37.8±1.36** |
| resnet50 Engstrom et al. [56] | 63.4 | 27.6 | SA | 59.8±0.64 | 57.2±0.79 | 52.9±1.11 | 48.6±1.31 |
| | | | AA | 54.6±0.99 | 52.8±1.09 | 50.3±1.43 | 48.1±1.18 |
| | | | MSA | **52.5±1.23** | **50.8±1.47** | **48.0±1.15** | **45.8±1.35** |

Table 5: MSA trained on a CIFAR10 model attacking the **undefended ImageNet models** in the $\ell_\infty$ threat model with $\epsilon = 0.05$. We compare our method with SA and set $p^0 = 0.05$ for the untargeted case and $p^0 = 0.01$ for the targeted case as suggested by Andriushchenko et al. [1]. For the targeted attacks robust accuracy is the fraction of the total number of images that was initially correctly classified by the model and not shifted to the target class during the attack. We provide clean accuracy of the models on a subset of 1000 ImageNet validation set images that we consider.

| Model | Clean acc. (%) | Attack | Untargeted | | | | Targeted | | | |
|---|---|---|---|---|---|---|---|---|---|---|
| | | | 500 | 1000 | 2500 | 5000 | 500 | 1000 | 2500 | 5000 |
| ResNet-50 [58] | 77.3 | SA | 8.8 | 5.1 | 0.2 | **0.0** | 76.9 | 75.1 | 62.5 | 34.4 |
| | | MSA | **2.9** | **0.8** | **0.0** | **0.0** | **67.1** | **52.0** | **27.8** | **12.1** |
| VGG-16-BN [59] | 75.0 | SA | 2.8 | 0.9 | **0.0** | **0.0** | 74.5 | 72.2 | 51.5 | 17.4 |
| | | MSA | **1.8** | **0.2** | **0.0** | **0.0** | **62.5** | **45.2** | **16.5** | **3.5** |
| Inception v3 [60] | 77.6 | SA | 16.6 | 6.1 | **2.3** | **1.0** | 77.5 | 76.4 | 70.9 | 59.8 |
| | | MSA | **10.2** | **5.1** | 2.6 | 1.3 | **74.4** | **70.6** | **60.1** | **49.5** |

Table 6: $\ell_2$-**threat model:** $\mathrm{MSA}_2$ uses the update size controller trained for the $\ell_2$ attack on a CIFAR10 model (see Section 4.1). The color controller is the same as for the $\ell_\infty$ case. $\ell_2$-MSA outperforms consistently $\ell_2$-SA [1] and $\ell_2$-AA [2] in terms of $\ell_2$-robust accuracy for $\epsilon = 0.5$ on 1000 CIFAR10 images. The full results for 5 models can be found in the Table 10.

| Model | Accuracy (%) | | Attack | Query budget | | | |
|---|---|---|---|---|---|---|---|
| | Clean | Robust | | 500 | 1000 | 2500 | 5000 |
| Ding et al. [3] | 88.02 | 66.09 | SA | 85.5±0.06 | 83.9±0.18 | 81.1±0.00 | 78.7±0.06 |
| | | | AA | 82.8±0.09 | 81.4±0.06 | 79.3±0.07 | 77.6±0.09 |
| | | | $\mathrm{MSA}_2$ | **82.3±0.03** | **80.9±0.07** | **77.4±0.09** | **75.8±0.19** |
| Rice et al. [61] | 88.67 | 67.68 | SA | 86.3±0.07 | 84.7±0.10 | 81.4±0.26 | 79.7±0.07 |
| | | | AA | 83.7±0.12 | 81.4±0.09 | 79.9±0.09 | 78.6±0.18 |
| | | | $\mathrm{MSA}_2$ | **82.6±0.09** | **81.0±0.07** | **78.7±0.03** | **76.9±0.25** |

Figure 2a illustrates the behaviour of the update size controller $\mathrm{MSA}_s$. It shows how the controller chooses the update size over time in an artificial scenario where the success probability $P(r^t = 1)$ is modeled to be constant. As implemented also by the heuristic schedules SA and AA, update size decays over time. However, the decay pattern depends heavily on $P(r^t = 1)$, with slower decay for larger $P(r^t = 1)$. This property is not implemented by heuristic schedules, but makes sense intuitively: high $P(r^t = 1)$ corresponds to a situation where the current perturbation $\xi^t$ can be improved relatively easily by coarse-grained changes implemented by the current update sizes; in this case it makes sense to first get the coarse-grained structure "right", before proceeding to fine-grained details that can be captured by small squares.

Figure 2b illustrates the empirical behavior of the color controller $\mathrm{MSA}_c$ when attacking the model by Ding et al. [3]. Shown are histograms over 500 images for the frequency of specific colors being sampled up to the respective iteration. Prior work like SA and AA maintained a uniform distribution of colors. However, the learned controller shows a clear preference for sampling black and white more often than uniform ($p \approx 0.18$ vs. $p = 0.125$ for uniform), blue and yellow approximately with $p = 0.125$, and the other colors less often than uniform. Since the color controller depends on the success rates $R^t$ of colors, this behavior is not hard-coded into the controller but identified on-the-fly during the attack (so behavior can differ for models with different vulnerabilities).

Table 7 demonstrates the ablation studies with respect to the used controllers on the model by Gowal et al. [62]. We observe that $\mathrm{MSA}_s$ combined with the uniform color distribution alone significantly improves the results of SA and AA in most of the query budgets (with the exception of the strong extrapolation regime of 5000 queries). The controller $\mathrm{MSA}_c$ combined with all schedules improves the performance with the exception of 500 queries regime for SA and $\mathrm{MSA}_s$ where it provides an equal or a slightly worse result in some cases.

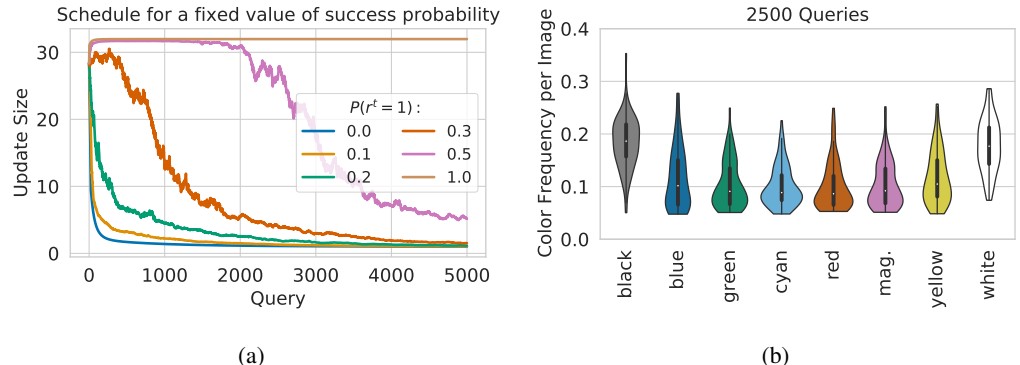

(a)                                                    (b)

Figure 2: (a) Square-size schedule obtained from $\text{MSA}_s$ over time for fixed values of success probability $P(r^t = 1)$ (averaged over 25 runs) (b) Illustration of color frequency histogram of 500 images after 2500 queries of $\text{MSA}_c$.

Table 7: We compare the update size controller $\text{MSA}_s$ to the schedules from SA [1] and AA [2] in the $\ell_\infty$ threat model with $\epsilon = 8/255$ on a model by Gowal et al. [62] with 1000 CIFAR10 test images. We also compare the uniform color sampling to our color controller $\text{MSA}_c$. Averaged across 5 runs with different random seeds.

| Update size schedule | Color sampling | Query budget | | | |
|---|---|---|---|---|---|
| | | 500 | 1000 | 2500 | 5000 |
| SA | Uniform | 80.6±0.09 | 76.7±0.07 | 70.8±0.14 | 67.5±0.07 |
| SA | $\text{MSA}_c$ | 81.0±0.05 | 76.4±0.08 | 69.9±0.10 | **67.2±0.07** |
| AA | Uniform | 80.0±0.17 | 76.8±0.11 | 72.2±0.10 | 69.2±0.08 |
| AA | $\text{MSA}_c$ | 79.9±0.12 | 76.7±0.03 | 71.6±0.17 | 68.8±0.07 |
| $\text{MSA}_s$ | Uniform | **76.9±0.05** | 73.7±0.05 | 69.8±0.13 | 67.6±0.04 |
| $\text{MSA}_s$ | $\text{MSA}_c$ | **76.9±0.07** | **73.4±0.13** | **69.0±0.08** | **67.2±0.04** |

## 5   Conclusion

In this work we propose a theoretical framework for meta-learning search distributions that help to improve efficiency of random search based black-box adversarial attacks. We implement and investigate this framework for Square Attack with $\ell_\infty$ and $\ell_2$ perturbations. Our experimental results show that learned adaptive controllers improve attack performance across different query budgets and generalize to new datasets as well as targeted attacks. Future directions are applying our framework to other random-search based attacks and threat models as well as learning controllers for sampling positions or even geometric primitives (going beyond squares).

**Ethical and Societal Impact**   This work contributes to the field of black-box adversarial attacks, which can be used for benign purposes like reliably evaluating the robustness of ML systems as well as malign ones such as identifying and exploiting weaknesses of these systems. Increasing the query-efficiency and success rate of black-box attacks can amplify safety and security concerns in domains such as highly automated driving or robotics. Meta Square Attack allows more efficient generation of imperceptible image distortions that may allow bypassing automated content-control systems for filtering images with violent, pornographic, or otherwise offensive content. We are optimistic that the research on adversarial attacks contributes to developing defenses and mitigation strategies that outweigh these risks in the medium term. A general mitigation strategy could be to add a check to a system which rejects predictions for a sequence of inputs which all differ by at most an $\ell_p$ distance of some $\epsilon$. This would make most score-based black-box attacks for image-specific perturbations ineffective, including Meta Square Attack.

## Acknowledgements

Matthias Hein is a member of the Machine Learning Cluster of Excellence, EXC number 2064/1 – Project number 390727645 and of the BMBF Tübingen AI Center, FKZ: 01IS18039B.

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
