# A Appendix

## A.1 Full experimental results

In this section we provide the full experimental results that extend the results demonstrated in the Section 4.2. Table 8 demonstrates the evaluation on 16 robustly trained CIFAR10 models from RobustBench [28] that was summarized in the Table 2. We consider four configurations of the attack for each of the models. SA and AA correspond to the update size schedules proposed by Andriushchenko et al. [1] and Croce and Hein [2] respectively. "Uni" denotes sampling the color for the update uniformly. $MSA_s$+$MSA_c$ is a combination of an update size controller with the color sampling controller that we denoted as MSA in the Section 4.2. $MSA_s$+Uni is an ablated version in which we only use an update size controller $MSA_s$. The clean and robust accuracy of the models are taken from `https://robustbench.github.io/`.

Table 8: We compare the update size controller $MSA_s$ to the schedules from the SA [1] and the AA [2] in the $\ell_\infty$ threat model with $\epsilon = 8/255$ on 1000 CIFAR10 test images. We also compare the uniform color sampling denoted as "Uni" to our color controller $MSA_c$. Averaged across at least 3 runs with different random seeds.

| Model | Accuracy (%) | | Square size | Color | Query budget | | | |
|---|---|---|---|---|---|---|---|---|
| | Clean | Robust | | | 500 | 1000 | 2500 | 5000 |
| Wong et al. [51] | 83.34 | 43.21 | SA | Uni | 69.7±0.15 | 63.5±0.10 | 55.1±0.04 | 50.8±0.08 |
| | | | AA | Uni | 69.5±0.21 | 63.9±0.10 | 57.4±0.07 | 53.6±0.06 |
| | | | $MSA_s$ | Uni | **63.9±0.11** | 59.8±0.10 | 54.0±0.16 | 51.1±0.08 |
| | | | $MSA_s$ | $MSA_c$ | **63.9±0.12** | **59.1±0.09** | **53.0±0.16** | **49.8±0.08** |
| Ding et al. [3] | 84.36 | 41.44 | SA | Uni | 68.7±0.20 | 63.2±0.28 | 57.8±0.13 | 54.9±0.17 |
| | | | AA | Uni | 66.6±0.18 | 62.2±0.14 | 57.5±0.12 | 55.0±0.20 |
| | | | $MSA_s$ | Uni | 62.4±0.15 | 59.4±0.09 | 56.1±0.10 | 54.6±0.06 |
| | | | $MSA_s$ | $MSA_c$ | **62.2±0.14** | **59.1±0.16** | **55.9±0.15** | **54.1±0.15** |
| Engstrom et al. [56] | 87.03 | 49.25 | SA | Uni | 72.8±0.19 | 67.4±0.21 | 59.9±0.17 | 56.3±0.07 |
| | | | AA | Uni | 71.9±0.1 | 67.9±0.14 | 61.6±0.12 | 58.0±0.06 |
| | | | $MSA_s$ | Uni | 67.9±0.12 | 64.2±0.15 | 58.9±0.05 | 56.4±0.12 |
| | | | $MSA_s$ | $MSA_c$ | **67.8±0.12** | **63.4±0.18** | **58.2±0.06** | **55.9±0.04** |
| Gowal et al. [62] | 89.48 | 62.76 | SA | Uni | 80.6±0.09 | 76.7±0.06 | 70.8±0.14 | 67.5±0.07 |
| | | | AA | Uni | 80.0±0.17 | 76.8±0.11 | 72.2±0.10 | 69.2±0.08 |
| | | | $MSA_s$ | Uni | **76.9±0.05** | 73.7±0.05 | 69.8±0.13 | 67.6±0.04 |
| | | | $MSA_s$ | $MSA_c$ | **76.9±0.07** | **73.4±0.13** | **69.0±0.08** | **67.2±0.04** |
| Carmon et al. [63] | 89.69 | 59.53 | SA | Uni | 79.0±0.15 | 76.0±0.14 | 68.2±0.07 | 65.4±0.09 |
| | | | AA | Uni | 78.0±0.10 | 74.5±0.11 | 69.6±0.04 | 67.1±0.05 |
| | | | $MSA_s$ | Uni | **74.4±0.09** | 70.8±0.06 | 67.5±0.07 | 65.6±0.07 |
| | | | $MSA_s$ | $MSA_c$ | 74.6±0.10 | **70.3±0.07** | **67.0±0.07** | 65.4±0.08 |
| Huang et al. [52] | 83.48 | 53.34 | SA | Uni | 72.3±0.1 | 66.6±0.16 | 60.5±0.09 | 57.3±0.08 |
| | | | AA | Uni | 70.6±0.10 | 66.5±0.07 | 61.2±0.10 | 58.5±0.11 |
| | | | $MSA_s$ | Uni | 66.4±0.12 | 63.4±0.08 | 59.2±0.07 | 57.4±0.15 |
| | | | $MSA_s$ | $MSA_c$ | **66.1±0.10** | **62.9±0.15** | **58.7±0.05** | **56.8±0.08** |
| Andriushchenko and Flammarion [64] | 79.84 | 43.93 | SA | Uni | 66.0±0.22 | 60.5±0.24 | 54.0±0.06 | 50.2±0.03 |
| | | | AA | Uni | 64.6±0.12 | 60.2±0.22 | 55.7±0.14 | 52.1±0.15 |
| | | | $MSA_s$ | Uni | 60.4±0.09 | 57.0±0.07 | 52.5±0.19 | 50.0±0.06 |
| | | | $MSA_s$ | $MSA_c$ | **60.1±0.07** | **56.8±0.15** | **51.9±0.15** | **49.4±0.22** |
| Zhang et al. [65] | 84.92 | 53.08 | SA | Uni | 72.3±0.03 | 67.2±0.19 | 62.0±0.09 | 59.0±0.06 |
| | | | AA | Uni | 70.8±0.22 | 67.2±0.17 | 62.7±0.10 | 60.3±0.17 |
| | | | $MSA_s$ | Uni | 67.5±0.06 | 64.2±0.18 | 60.8±0.07 | 59.0±0.07 |
| | | | $MSA_s$ | $MSA_c$ | **66.8±0.09** | **63.9±0.07** | **60.4±0.06** | **58.7±0.13** |
| Hendrycks et al. [66] | 87.11 | 54.92 | SA | Uni | 75.3±0.30 | 69.8±0.19 | 64.2±0.15 | 60.8±0.00 |
| | | | AA | Uni | 74.7±0.17 | 70.5±0.26 | 64.7±0.12 | 62.8±0.15 |
| | | | $MSA_s$ | Uni | 71.1±0.07 | 66.6±0.15 | 63.2±0.12 | 61.0±0.07 |
| | | | $MSA_s$ | $MSA_c$ | **70.6±0.17** | **66.1±0.12** | **62.7±0.13** | **60.4±0.15** |
| Wang et al. [67] | 87.50 | 56.29 | SA | Uni | 77.7±0.12 | 72.2±0.03 | 65.8±0.15 | 62.2±0.03 |
| | | | AA | Uni | 76.7±0.06 | 72.8±0.12 | 67.6±0.20 | 64.0±0.17 |
| | | | $MSA_s$ | Uni | 73.1±0.18 | 69.8±0.09 | 64.9±0.15 | 62.3±0.03 |
| | | | $MSA_s$ | $MSA_c$ | **72.7±0.07** | **69.5±0.09** | **64.3±0.18** | **62.0±0.07** |
| Cui et al. [54] | 88.22 | 52.86 | SA | Uni | 75.5±0.22 | 69.6±0.09 | 62.9±0.20 | 59.2±0.15 |
| | | | AA | Uni | 74.2±0.13 | 70.2±0.07 | 64.8±0.07 | 61.1±0.23 |
| | | | $MSA_s$ | Uni | 70.1±0.07 | 66.7±0.18 | 61.8±0.12 | 59.7±0.03 |
| | | | $MSA_s$ | $MSA_c$ | **70.0±0.15** | **66.2±0.25** | **60.8±0.09** | **59.0±0.06** |

| Model | Accuracy (%) | | Square size | Color | Query budget | | | |
|---|---|---|---|---|---|---|---|---|
| | Clean | Robust | | | 500 | 1000 | 2500 | 5000 |
| Sitawarin et al. [68] | 86.84 | 50.72 | SA | Uni | 73.4±0.06 | 66.4±0.10 | 61.1±0.07 | 57.4±0.12 |
| | | | AA | Uni | 72.0±0.20 | 66.8±0.23 | 62.3±0.20 | 59.4±0.12 |
| | | | $MSA_s$ | Uni | **66.7±0.06** | 63.6±0.03 | 60.3±0.17 | 57.5±0.03 |
| | | | $MSA_s$ | $MSA_c$ | 66.9±0.00 | **63.1±0.09** | **59.3±0.12** | **57.0±0.00** |
| Wu et al. [53] | 85.36 | 56.17 | SA | Uni | 75.0±0.19 | 69.7±0.21 | 63.8±0.12 | 60.4±0.07 |
| | | | AA | Uni | 73.6±0.03 | 69.5±0.25 | 64.5±0.07 | 62.3±0.07 |
| | | | $MSA_s$ | Uni | 69.6±0.09 | 66.1±0.20 | 63.1±0.17 | 60.7±0.07 |
| | | | $MSA_s$ | $MSA_c$ | **69.4±0.23** | **65.7±0.12** | **62.6±0.12** | **60.3±0.03** |
| Zhang et al. [69] | 89.36 | 59.64 | SA | Uni | 79.6±0.27 | 74.6±0.03 | 66.9±0.07 | 64.0±0.07 |
| | | | AA | Uni | 78.4±0.06 | 75.3±0.03 | 68.9±0.06 | 65.6±0.03 |
| | | | $MSA_s$ | Uni | 75.1±0.09 | 71.4±0.09 | 66.2±0.20 | 64.3±0.06 |
| | | | $MSA_s$ | $MSA_c$ | **75.0±0.19** | **70.4±0.17** | **65.6±0.09** | **63.8±0.10** |
| Zhang et al. [70] | 84.52 | 53.51 | SA | Uni | 73.4±0.03 | 67.5±0.09 | 61.5±0.12 | **58.9±0.09** |
| | | | AA | Uni | 72.3±0.06 | 67.7±0.00 | 62.3±0.06 | 60.4±0.06 |
| | | | $MSA_s$ | Uni | **67.4±0.09** | 63.6±0.25 | 61.2±0.03 | 59.3±0.10 |
| | | | $MSA_s$ | $MSA_c$ | 67.6±0.06 | **63.5±0.09** | **60.6±0.10** | 59.0±0.10 |
| Zhang et al. [71] | 87.20 | 44.83 | SA | Uni | 73.1±0.00 | 66.2±0.26 | 56.5±0.15 | 52.5±0.12 |
| | | | AA | Uni | 71.8±0.23 | 66.5±0.07 | 59.2±0.09 | 54.7±0.12 |
| | | | $MSA_s$ | Uni | 66.9±0.18 | 61.9±0.09 | 55.2±0.15 | 52.6±0.09 |
| | | | $MSA_s$ | $MSA_c$ | **66.4±0.06** | **60.8±0.15** | **54.6±0.09** | **51.9±0.12** |

Table 9 provides an extended version of the Table 4 in which we additionally provide the results for the ablated version $MSA_s$+Uni to demonstrate that the update size controller on its own provides better results than the considered baselines SA [1] and AA [2]. However, if we add a color sampling controller $MSA_c$, we manage to further improve the robust accuracy estimate.

Table 10 demonstrates the the results for the $\ell_2$ threat model for five $\ell_2$ robust models from Robust-Bench [28]. We have chosen the models for which Croce and Hein [2] provide their evaluation of the Square Attack. They evaluate on the whole CIFAR10 test set and we evaluate on a subset of 1000 test images. Therefore their estimate is not identical to our entry AA+Uni. But we still provide it as Sq AA [2] in the Table 10 for additional reference.

## A.2 Meta-training the Controllers

The meta-training of controllers was described in Section 3 and Section 4.1. We summarize it schematically for the case of general random-search based black-box attack (Figure 3) and for the Meta Square Attack (Figure 4). We provide some additional details and illustrate the learning curves.

Table 9: Results of attacking 1000 ImageNet validation set images with $\ell_\infty$ threat model and $\epsilon = 4/255$ as in Croce and Hein [2]. For the SA update size schedule, we use the parameter $p^0 = 0.05$ as suggested in Andriushchenko et al. [1]. AA and Uni are defined as in Table 1. $MSA_s$ and $MSA_c$ are meta-trained on CIFAR10 (see Section 4.1 for details). We report mean and standard error of robust accuracy for different queries budgets across 3 runs with different random seeds.

| Model | Accuracy (%) | | Square size | Color | Query budget | | | |
|---|---|---|---|---|---|---|---|---|
| | Clean | Robust | | | 500 | 1000 | 2500 | 5000 |
| resnet18 Salman et al. [55] | 52.5 | 25.0 | SA | Uni | 50.6±1.43 | 48.1±1.18 | 43.9±1.00 | 40.3±1.21 |
| | | | AA | Uni | 45.2±1.09 | 43.5±0.86 | 41.0±1.07 | 39.0±1.21 |
| | | | $MSA_s$ | Uni | 43.4±0.94 | **41.7±1.13** | 39.5±1.07 | 38.3±1.33 |
| | | | $MSA_s$ | $MSA_c$ | **43.3±1.00** | **41.7±0.94** | **39.1±1.23** | **37.8±1.36** |
| resnet50 Engstrom et al. [56] | 63.4 | 27.6 | SA | Uni | 59.8±0.64 | 57.2±0.79 | 52.9±1.11 | 48.6±1.31 |
| | | | AA | Uni | 54.6±0.99 | 52.8±1.09 | 50.3±1.43 | 48.1±1.18 |
| | | | $MSA_s$ | Uni | 52.6±1.07 | 51.2±1.40 | 48.3±1.22 | **45.8±1.26** |
| | | | $MSA_s$ | $MSA_c$ | **52.5±1.23** | **50.8±1.47** | **48.0±1.15** | **45.8±1.35** |

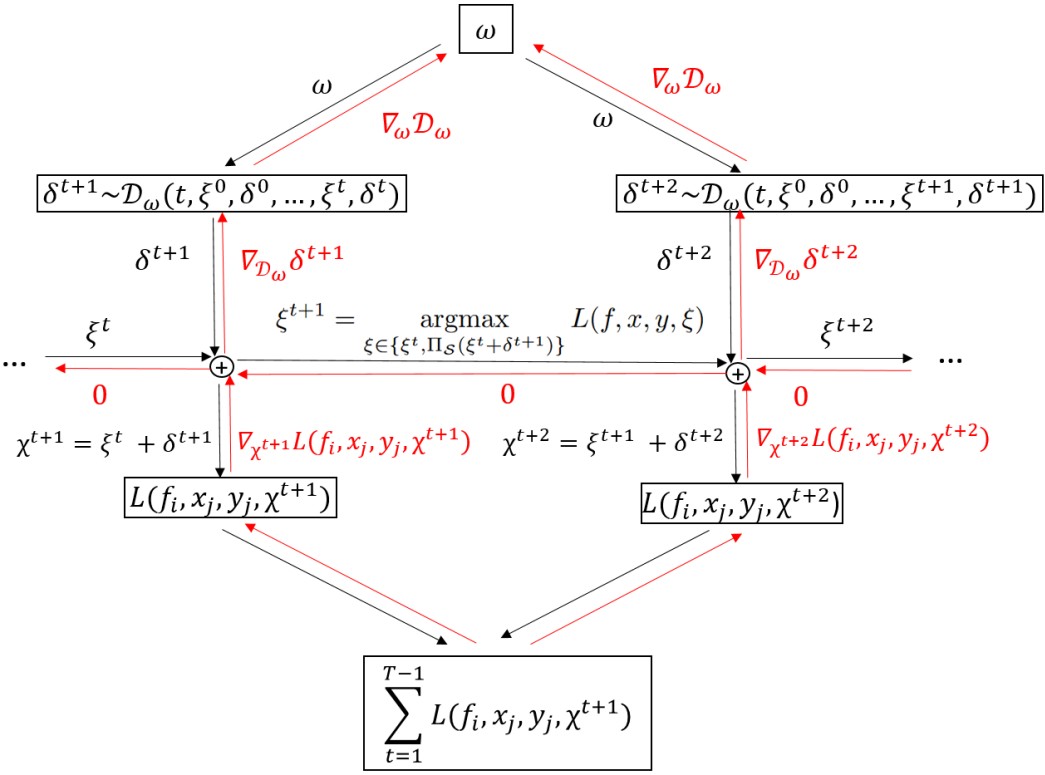

Figure 3: Schematic illustration of the general meta-learning procedure described in Section 3.2 for two subsequent steps of the random search attack (3). We describe forward (black) and backward (red) pass. $\nabla_x f$ denotes gradient for scalar-valued functions $f$ and Jacobian for vector-valued functions. Differential expressions with search distribution $\mathcal{D}_\omega$ are informal and in need to be handled with reparametrization trick or other methods when applying the method to particular attacks (for examples see Section 3.3). Please note that the gradient with respect to $\xi^t$ is set to 0 for all $t$ (see Section 3.2).

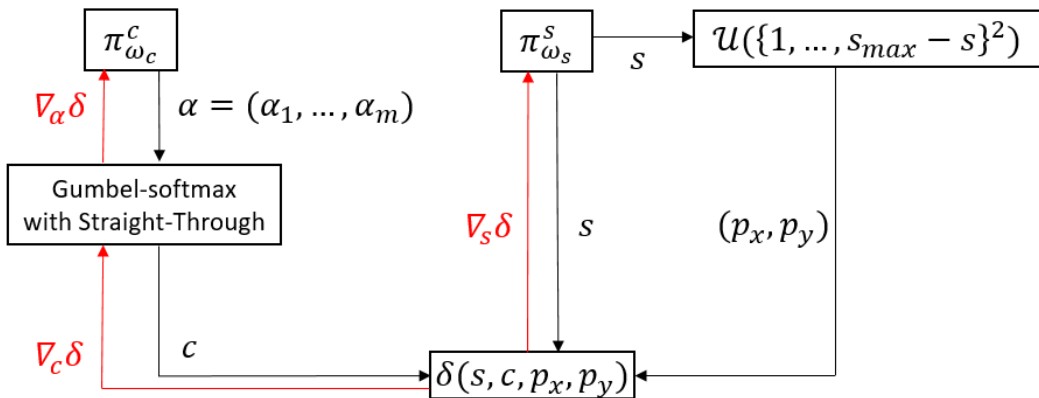

Figure 4: Illustration of Meta Square Attack described in Section 3.3. The search distribution $\mathcal{D}_{(s,c)}$ depends on parameters $s$ and $\alpha$ that are provided by the update size controller $\pi^c_{\omega_c}$ and the color controller $\pi^s_{\omega_s}$, respectively. Square positions $(p_x, p_y)$ are sampled from the uniform distribution $\mathcal{U}(\{1, ..., s_{max} - s\}^2)$

.

Table 10: $\mathrm{MSA}_s^2$ is the update size controller trained for the $\ell_2$ attack on a CIFAR10 model described in the Section 4.1. $\mathrm{MSA}_s^\infty$ denotes the update size controller meta-trained for the $\ell_\infty$ Square Attack on CIFAR10. The color controller $\mathrm{MSA}_c$ is the same as for the $\ell_\infty$ case. We compare to the $\ell_2$ versions of SA [1] and AA [2] with $\epsilon = 0.5$ on 1000 CIFAR10 images.

| Model | Accuracy (%) | | | Square size | Color | Query budget | | | |
|---|---|---|---|---|---|---|---|---|---|
| | Clean | Robust | Sq AA [2] | | | 500 | 1000 | 2500 | 5000 |
| Ding et al. [3] | 88.02 | 66.09 | 76.99 | SA | Uni | 85.5±0.06 | 83.9±0.08 | 81.1±0.00 | 78.7±0.06 |
| | | | | AA | Uni | 82.8±0.09 | 81.4±0.06 | 79.2±0.00 | 77.7±0.15 |
| | | | | $\mathrm{MSA}_s^\infty$ | Uni | 82.6±0.09 | 81.7±0.12 | 79.8±0.12 | 77.9±0.07 |
| | | | | $\mathrm{MSA}_s^\infty$ | $\mathrm{MSA}_c$ | 82.5±0.22 | 81.5±0.03 | 78.5±0.06 | 76.9±0.17 |
| | | | | AA | $\mathrm{MSA}_c$ | 82.6±0.03 | 81.1±0.17 | 78.2±0.10 | 76.5±0.09 |
| | | | | $\mathrm{MSA}_s^2$ | $\mathrm{MSA}_c$ | **82.3±0.03** | **80.9±0.07** | **77.4±0.09** | **75.8±0.19** |
| Rice et al. [61] | 88.67 | 67.68 | 79.01 | SA | Uni | 86.3±0.07 | 84.7±0.10 | 81.4±0.15 | 79.7±0.07 |
| | | | | AA | Uni | 83.7±0.12 | 81.4±0.09 | 79.9±0.09 | 78.6±0.18 |
| | | | | $\mathrm{MSA}_s^\infty$ | Uni | 83.2±0.12 | 81.8±0.07 | 80.1±0.07 | 79.1±0.09 |
| | | | | $\mathrm{MSA}_s^\infty$ | $\mathrm{MSA}_c$ | 83.0±0.15 | 81.2±0.06 | 79.6±0.03 | 78.3±0.03 |
| | | | | AA | $\mathrm{MSA}_c$ | 83.4±0.12 | 81.2±0.10 | 79.3±0.09 | 78.0±0.06 |
| | | | | $\mathrm{MSA}_s^2$ | $\mathrm{MSA}_c$ | **82.6±0.09** | **81.0±0.07** | **78.7±0.03** | **76.9±0.25** |
| Augustin et al. [72] | 91.08 | 72.91 | 83.10 | SA | Uni | 89.0 | 88.4 | 86.9 | 84.2 |
| | | | | AA | Uni | 87.8±0.03 | 86.8±0.09 | 84.8±0.17 | 83.3±0.17 |
| | | | | $\mathrm{MSA}_s^\infty$ | Uni | 87.7±0.06 | 87.0±0.09 | 85.2±0.03 | 83.4±0.10 |
| | | | | $\mathrm{MSA}_s^\infty$ | $\mathrm{MSA}_c$ | 87.4±0.15 | 86.5±0.12 | 84.1±0.20 | 82.8±0.13 |
| | | | | AA | $\mathrm{MSA}_c$ | 87.7±0.12 | 86.6±0.09 | 83.9±0.13 | 82.7±0.09 |
| | | | | $\mathrm{MSA}_s^2$ | $\mathrm{MSA}_c$ | **87.5±0.12** | **86.3±0.06** | **83.4±0.03** | **81.8±0.07** |
| Engstrom et al. [56] | 90.83 | 69.24 | 80.92 | SA | Uni | 87.3 | 86.1 | 84.0 | 80.8 |
| | | | | AA | Uni | 85.3±0.06 | 83.7±0.15 | 81.5±0.24 | 79.5±0.18 |
| | | | | $\mathrm{MSA}_s^\infty$ | Uni | 85.2±0.12 | 84.2±0.17 | 82.0±0.09 | 79.9±0.07 |
| | | | | $\mathrm{MSA}_s^\infty$ | $\mathrm{MSA}_c$ | 85.1±0.07 | 83.7±0.03 | 80.6±0.06 | 78.8±0.09 |
| | | | | AA | $\mathrm{MSA}_c$ | 85.2±0.07 | 83.5±0.07 | 80.6±0.06 | 78.5±0.15 |
| | | | | $\mathrm{MSA}_s^2$ | $\mathrm{MSA}_c$ | **84.7±0.09** | **83.1±0.06** | **79.7±0.13** | **77.4±0.00** |
| Rony et al. [73] | 89.05 | 66.44 | 78.05 | SA | Uni | 85.4 | 83.5 | 80.5 | 78.3 |
| | | | | AA | Uni | 82.0±0.10 | 80.8±0.10 | 78.9±0.03 | 77.0±0.15 |
| | | | | $\mathrm{MSA}_s^\infty$ | Uni | 81.8±0.03 | 81.0±0.10 | 79.1±0.15 | 77.7±0.07 |
| | | | | $\mathrm{MSA}_s^\infty$ | $\mathrm{MSA}_c$ | 81.9±0.07 | 80.7±0.06 | 78.5±0.17 | 76.5±0.03 |
| | | | | AA | $\mathrm{MSA}_c$ | 81.9±0.09 | 80.6±0.12 | 78.3±0.07 | 76.2±0.03 |
| | | | | $\mathrm{MSA}_s^2$ | $\mathrm{MSA}_c$ | **81.6±0.03** | **80.4±0.00** | **77.2±0.00** | **75.7±0.09** |

As discussed in Section 3.3, we maximize the following meta-objective:

$$R(F, D, \omega) = \frac{1}{T} \sum_{f_i} \sum_{(x_j, y_j)} \sum_{t=1}^{T-1} L(f_i, x_j, y_j, \Pi_{\mathcal{S}}(\xi^t + \delta^{t+1})), \qquad (6)$$

where $\delta^{t+1} \sim \mathcal{D}_\omega(t, \xi^0, \delta^0, \dots, \xi^t, \delta^t)$. Recall from Section 3.1 that $L(f, x, y, \xi) := l(f(a(x, \xi)), y)$. As discussed in in Section 3.3, we use total loss improvement over attack as our meta-loss. Therefore, we choose $l$ in a way that it represents loss improvement caused by the update $\xi$ i. e.

$$l(f(a(x, \xi)), y) = (h(f(a(x, \xi)), y) - h_{max})_+, \qquad (7)$$

where $(x)_+$ is positive part function, $h(p, q)$ is cross-entropy loss and $h_{max}$ is the largest cross-entropy value obtained so far. In our case at step $t$ we have $h_{max} = h(f(a(x, \Pi_{\mathcal{S}}(\xi^t)), y)$ by design of the random search attack (3). Finally, instead of solving the problem of maximizing $R(F, D, \omega)$, we are solving the equivalent problem of minimizing $-R(F, D, \omega)$. Therefore, the loss that we use for training our controllers for Meta Square Attack is:

$$R_{MSA}(F, D, \omega) = -\frac{1}{T} \sum_{f_i} \sum_{(x_j, y_j)} \sum_{t=1}^{T-1} (h(f_i(a(x_j, \Pi_{\mathcal{S}}(\xi^t + \delta^{t+1}))), y_j) - h(f_i(a(x_j, \Pi_{\mathcal{S}}(\xi^t)), y_j))_+.$$
$$(8)$$

As discussed in Section 4, we use 1000 CIFAR10 test set images for meta-training and different 1000 images for evaluation. We use the default order of CIFAR10 images (i. e., we do not shuffle).

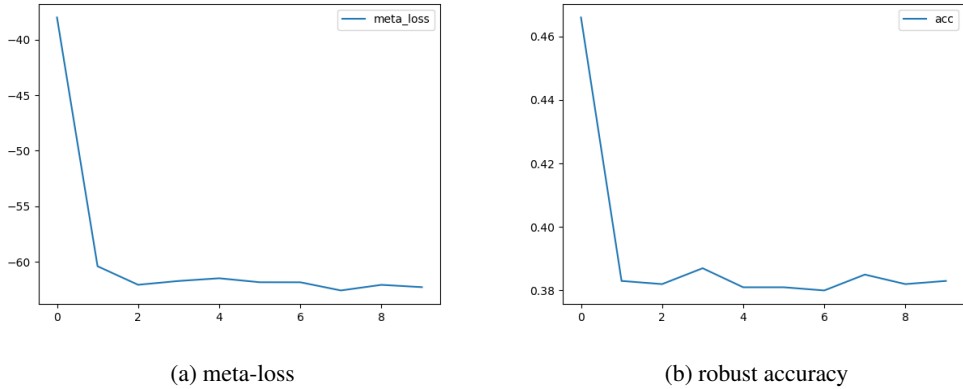

(a) meta-loss          (b) robust accuracy

Figure 5: Meta-loss and robust accuracy on the training set during meta-training.

For meta-training we use images from 0 to 999 and for evaluation we use images from 9000 to 9999. Figure 5 demonstrates the minimization of $R_{MSA}(F, D, \omega)$ and corresponding behavior of the accuracy on the training set. One can see that the proposed meta-loss $R_{MSA}(F, D, \omega)$ serves as a reasonable differentiable proxy for the robust accuracy. We observe that the loss reaches a close-to-minimal value already after two epochs.

### A.3 Square relaxation

In Section 3.3, we formalize update size and color controllers that we learn for Meta Square Attack. Here we provide additional details on how we avoid blocking of gradient flow in our optimization scheme using relaxed square sampling.

$$g = \frac{1}{T} \sum_{f_i} \sum_{(x_j, y_j)} \sum_{t=1}^{T-1} \nabla_\omega L(f_i, x_j, y_j, \xi^t + \delta^{t+1}), \tag{9}$$

for simplicity assuming that projection operator $\Pi_{\mathcal{S}}$ in Equation (5) is incorporated into $L$. Since we rewrite $\nabla_\omega L(f_i, x_j, y_j, \xi^t + \delta^{t+1}) = \nabla_{\delta^{t+1}} L(f_i, x_j, y_j, \xi^t + \delta^{t+1}) \nabla_\omega \delta^{t+1}$, we need to compute the Jacobian $\nabla_\omega \delta^{t+1}$ of update vector $\delta^{t+1}$ with respect to meta-parameters $\omega$.

Recall that in Section 3.3 we denote $\omega = (\omega_s, \omega_c)$ and consider controllers $\pi_{\omega_s}^s$ and $\pi_{\omega_c}^c$ for the update size and color respectively. Since computing $\nabla_{\omega_c} \delta^{t+1}$ is done via Gumbel softmax [46, 47], here we concentrate on computing $\nabla_{\omega_s} \delta^{t+1}$. Since $\pi_{\omega_s}^s$ only controls update size, we assume its position and color to be fixed when computing the gradient.

In SA [1] each update is parametrized by an integer square width from $\{1, ..., w\}$ where $w$ is image width. This parameter is obtained by rounding real value $s$ obtained from the update size schedule to the closest integer in the feasible range. During meta-training we cannot round the output $s$ of $\pi_{\omega_s}^s$ since in that case we get $\nabla_{\omega_s} \delta^{t+1} = 0$ almost everywhere. Therefore, we propose a differentiable relaxation (see Figure 6). The inner part of the square with width $odd(s) = 2 \cdot \lfloor \frac{s-1}{2} \rfloor + 1$ is filled with the sampled color $c$ completely. The color of pixels in the 1-pixel boundary is interpolated between the background color $c_0$ and the new color $c$ as: $k \cdot c + (1 - k) \cdot c_0$. The coefficient $k$ of the new color is equal to the fraction that the square of non-integer width $s$ would occupy in the respective pixel. Therefore, for the 4-neighborhood the new color fraction is $k = \frac{s - odd(s)}{2}$ and for the pixel of 8-neighborhood that do not belong to 4-neighborhood $k = (\frac{s - odd(s)}{2})^2$.

### A.4 Additional analysis of the learned controllers

In this Section we provide some additional analysis of the meta-learned controllers that we have started in the Section 4.3.

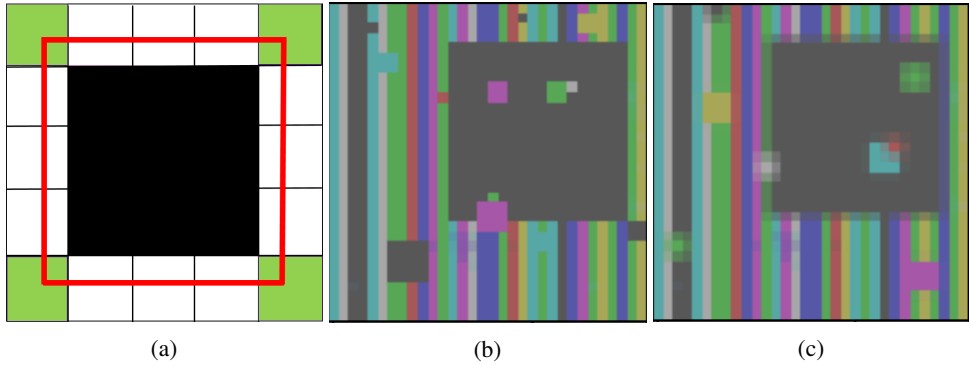

(a)                              (b)                              (c)

Figure 6: (a) illustration of a square with non-integer size $s$ (red), size $odd(s)$ (black), 4-neighborhood (white) and 8-neighborhood pixels that do not belong to 4-neighborhood (green), (b) standard square attack perturbation, (c) square attack perturbation with proposed square relaxation.

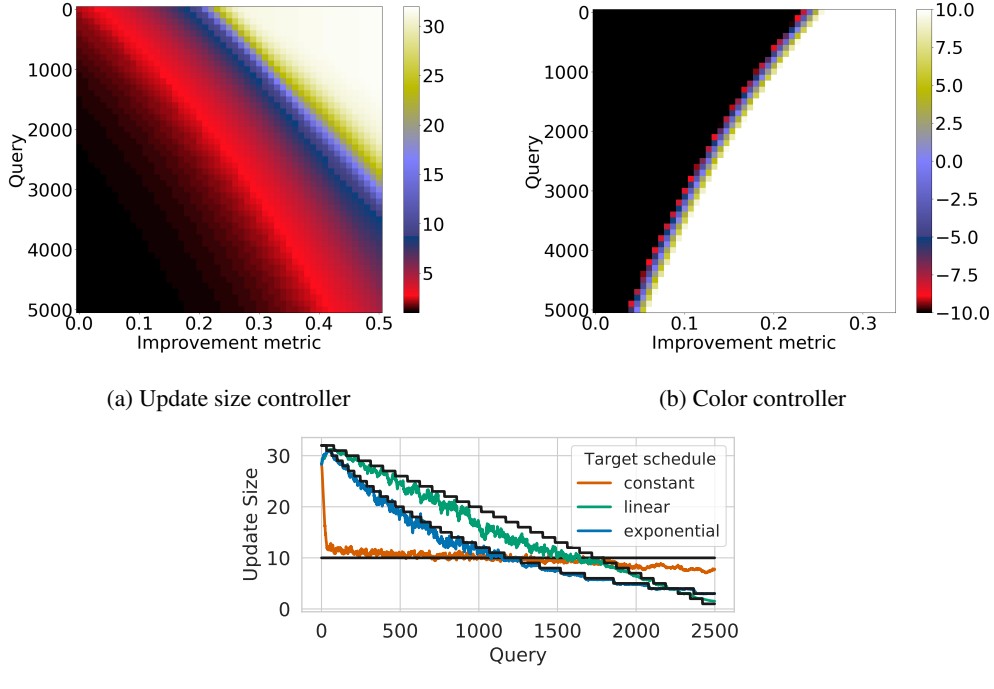

(a) Update size controller

(b) Color controller

(c) Update size schedule adjustment

Figure 7: Additional analysis of the meta-learned controllers. The plot of the (a) update size controller $\mathrm{MSA}_s$ and (b) color controller $\mathrm{MSA}_c$ as functions of their inputs. (c) $\mathrm{MSA}_s$ adjustment to the target schedules (averaged over 25 runs).

Since our controllers are functions of 2 inputs as described in the Section 3.3 we can illustrate the dependence of their outputs on these inputs. We show it in Figure 7a for the update size controller and Figure 7b for the color controller.

The Figure 7c illustrates observed schedules for idealized (and untypical) target schedules: these target schedules are unknown to the controller and are encoded in the success probabilities by setting $p(r^t = 1) = 0.4$ for update sizes smaller or equal to the value of the target schedules and to $p(r^t = 1) = 0.1$ otherwise. This abrupt change of the success probabilities and the shape of the target schedules "constant" and "linear" are very unlike the behavior of the attacks during meta-training; nevertheless the empirical schedules by the controller follow the target behavior reasonably good, indicating that the learned square-size controller generalizes well.

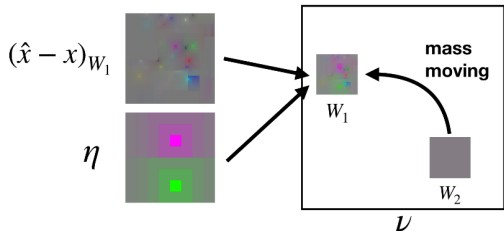

Figure 8: Perturbation of the $\ell_2$ attack. Image source: Andriushchenko et al. [1]

## A.5  Meta Square Attack for the $\ell_2$ threat model

To meta-learn the update size controller for the $\ell_2$ threat model we use the same procedure as discussed in Section 3.3. The only difference is the relaxation that we use to sample continous updates since the update geometry is different. See Section A.3 for the $\ell_\infty$ case.

The sampling procedure of the $\ell_2$ Square Attack is described in detail in the Algorithm 3 in Andriushchenko et al. [1]. On a high level the algoritm consists of 2 steps (Figure 8):

1. Take the mass from $W_2$
2. Update $W_1$

Let $s$ be non-integer square size. $odd(s)$ – the largest odd integer number not exceeding $s$, $odd(s) = 2 \cdot \lfloor \frac{s-1}{2} \rfloor + 1$. The performed update is a linear interpolation between the squares of size $odd(s)$ and $odd(s) + 2$. We denote $frac(s) = \frac{s-odd(s)}{2} \in [0;1)$ that will be an interpolation coefficient.

For the step 1 we consider the window $W_2$ of size $odd(s)$ and denote it's 1-pixel outer boundary as $W_2^B$. As in SA [1], we set the whole $W_2$ to 0 and add $||W_2||_2$ to the update budget. We also add $frac(s) \cdot ||W_2^B||_2$ to the budget, therefore taking $frac(s)$ part of the norm. We update the boundary as $W_{2,new}^B := \sqrt{1 - frac(s)^2} \cdot W_2^B$. We get $||W_2^B||_2^2 = ||W_{2,new}^B||_2^2 + frac(s)^2 \cdot ||W_2^B||_2^2$.

## A.6 Used data

In this work we only use the data published under formal licenses. To the best of our knowledge, data used in this project do not contain any personally identifiable information or offensive content.

For the CIFAR10 and CIFAR100 experiments in Table 8, we use pre-trained models from the RobustBench [28]. Information about architecture of the models and licenses of the corresponding model weights are in Table 11. Full texts of the licenses are available under the following link: `https://github.com/RobustBench/robustbench/blob/master/LICENSE`.

Table 11: Architecture and licenses of the models used in this work

| Dataset | Model | Architecture | Model weights license |
|---------|-------|--------------|----------------------|
| CIFAR10 | Wong et al. [51] | ResNet-18 | MIT |
| | Ding et al. [3] | WideResNet-28-4 | Attribution-NonCommercial-ShareAlike 4.0 International; Copyright (c) 2020, Borealis AI |
| | Engstrom et al. [56] | ResNet-50 | MIT |
| | Gowal et al. [62] | WideResNet-28-10 | Apache License 2.0; Copyright (c) 2021, Google |
| | Carmon et al. [63] | WideResNet-28-10 | MIT |
| | Huang et al. [52] | WideResNet-34-10 | MIT |
| | Andriushchenko and Flammarion [64] | PreActResNet-18 | MIT |
| | Zhang et al. [65] | WideResNet-34-10 | MIT |
| | Hendrycks et al. [66] | WideResNet-28-10 | Apache License 2.0; Copyright (c) 2019, Dan Hendrycks |
| | Wang et al. [67] | WideResNet-28-10 | MIT |
| | Cui et al. [54] | WideResNet-34-10 | MIT |
| | Sitawarin et al. [68] | WideResNet-34-10 | MIT |
| | Wu et al. [53] | WideResNet-34-10 | MIT |
| | Zhang et al. [69] | WideResNet-28-10 | MIT |
| | Zhang et al. [70] | WideResNet-34-10 | MIT |
| | Zhang et al. [71] | WideResNet-34-10 | MIT |
| | Rice et al. [61] | PreActResNet-18 | MIT |
| | Augustin et al. [72] | ResNet-50 | MIT |
| | Rony et al. [73] | WideResNet-28-10 | BSD 3-Clause License; Copyright (c) 2018, Jerome Rony |
| CIFAR100 | Wu et al. [53] | WideResNet-34-10 | MIT |
| | Cui et al. [54] | WideResNet-34-10 | MIT |
| ImageNet | Salman et al. [55] | ResNet-18 | MIT |
| | Engstrom et al. [56] | ResNet-50 | MIT |
| | He et al. [58] | ResNet-50 | BSD-3-Clause License (torchvision [74]) |
| | Simonyan and Zisserman [59] | VGG16-BN | BSD-3-Clause License (torchvision [74]) |
| | Szegedy et al. [60] | Inception v3 | BSD-3-Clause License (torchvision [74]) |