# OpenReview forum: "Meta-Learning the Search Distribution of Black-Box Random Search Based Adversarial Attacks"
_NeurIPS.cc/2021/Conference — NeurIPS 2021 Poster_

### Official Review · Reviewer_bYZY · 2021-07-06

**Rating:** 5
**Confidence:** 3

**Summary:**

The paper proposes a novel technique to improve query efficiency of black-box random search based adversarial attacks. The proposed method utilizes a controller which manages scheduling and proposal distribution for random search. The controller is trained by gradient-based meta-learning on surrogate models. Empirical results show that the Square Attack with the proposed method improves query efficiency compared to the original setting on CIFAR10 without manual fine-tuning.

**Limitations And Societal Impact:**

The authors have described limitations about the usage of surrogate models.

The authors have addressed societal impacts in Section 5.


**Main Review:**

Pros:
- The paper is well motivated.
- The characteristics of learned controller are useful and interesting.

Cons:
- The advantage over existing hyperparameter search methods is not clear.
- The proposed method needs the task specific controller design.
- Additional data and training are a strong assumption but not well explained.
- Experiments are not held in various settings.

While the motivation is clear and some of the results are interesting, it seems to need more explanations and experiments for the following points:

- The proposed method in this paper is similar to hyperparameter search (optimization) methods by Bayesian optimization or meta-learning, but the relationship with these methods is not explained. When random search based attacks only include simple hyperparameters like step size, these methods can also be used. In this case, does the proposed method have advantages over existing methods?
In addition, existing hyperparameter search methods would be applicable to the Square Attack under assumptions of simple distributions (for example, linear or quadratic size schedule). There is no empirical comparison with hyperparameter search methods, and it is not clear that the proposed method is better than the existing methods.

- The method needs the controller design which is specific to the task. The controller in Section 3.3 seems carefully designed and not simple. While the proposed method can be used without manual fine-tuning, the complex hand-designed architecture makes the advantage diminishing.

- The requirement of source models (i.e., surrogate models) and additional training is a strong assumption, and the assumption should be more explained. Although the authors explain this point by “on model with white-box access” in the abstract and introduction, the relationship between the white-box model and black-box model is not clear. In addition, while there are many existing methods which exploit surrogate models to improve the query efficiency, the paper does not mention the existing methods.

- In the experiment, the paper uses an adversarially trained ResNet model as source model, and all the target models are also adversarially trained ResNet-based models. This setting is not enough to show the strength of the proposed method. As the authors mentioned in line 66, the similarity between source and target models often affects the attack performance when the attack exploits source (surrogate) models. When the source model is trained with different architectures/training settings (e.g., DenseNet and naturally trained model), does the proposed method still outperform the baseline?

- In addition to the above similarity between the source model and target models, the experiments are held in only 1 dataset and 1 attack. Moreover, the improvement is not significant and there are still non-negligible margins between the result and robust accuracy. Therefore, it seems to need more experiments to show the advantage of the proposed method.


**Time Spent Reviewing:**

4.5 hours

---

> ### Author Response · Authors · 2021-08-10
> **Addressing the mentioned cons**
>
> We are grateful for the time devoted to reviewing the paper and the provided comments. We provide our responses to the open points below:
>
> - "Relation to hyperparameter search and Bayesian optimization"
>
> In our experiments we have observed that a fixed set of hyperparameters may perform well in one setting  but much worse in another. In particular, among the two proposed fixed schedules for the Square Attack none is dominating the other in all the considered query budget regimes (see Figure 1, Table 1). Achieving best possible performance under various query budgets is a crucial attack aspect in real-world security applications. Since using hyperparameter search or Bayesian Optimization for each particular query budget might be prohibitively expensive (or even impossible if we do not resort to a simple distribution of hyperparameters), we see one possible way in adjusting the parameters of the attack online. That is why we use the controllers to learn the adaptation principle rather than a set of hyperparameters.
>
> - "Controller design is specific to the task"
>
> As discussed in the Section 4.1, the controller itself, which is an MLP with 2 layers and 10 hidden units each, is quite simple and was purposefully not fine-tuned. The design of the controllers' inputs discussed in Section 3.3 (that might seem non-trivial) allows the inputs to be in the [0, 1] range which enhances the end-to-end training. The ability of this input form to generalize to the other settings is supported by the ImageNet transfer results in the Table 2 in the Appendix as well as the results provided in the Table R1 (the transfer of the controllers to CIFAR100), Table R2 (applying the method for the Square Attack with the $\ell_2$ threat model) R3 and Table R3 (the transfer of the controllers to the case of the targeted attacks).
>
> - "Relationship between the source model and the target model"
>
> Our goal was to demonstrate that although recently proposed state-of-the-art robust models use different training and augmentation techniques to improve robustness, the query efficiency of the Square Attack on these models can be uniformly improved by the controllers trained on a relatively simple robust source model. Therefore, as discussed in Section 4.1, we chose a source model trained with standard adversarial training so that it can be easily and cheaply acquired by the attacker. Only one of the models used in evaluation has the same ResNet18 architecture as the source model as can be seen in Table 3 in the Appendix.
>
> To address the rest of the points opened by the reviewer concerning the similarity between the source and the target models and testing the method for different datasets and attacks, we would like to refer to the experimental results provided in the Tables R1, R2, R3.
>
> The values provided as "Robust Accuracy" in Tables 1, 2 in the paper and Tables R1, R2 were obtained by using an ensemble of white-box and black-box attacks [1] and are given for reference. In this paper we have only black-box access to the network, which makes attacks significantly more difficult and thus one cannot expect to achieve the same level as white-box attacks.
>
> [1] F. Croce and M. Hein. Reliable evaluation of adversarial robustness with an ensemble of diverse parameter-free attacks. In ICML, 2020.

---

> > ### Comment · Reviewer_bYZY · 2021-08-23
> > **Post rebuttal comments**
> >
> > Thank you for the response. I understand the relation to hyperparameter search and Bayesian optimization. Additional experiments against undefended models in comments to other reviewers are meaningful. However, there are still some concerns. As a result, I will raise my score to 5.
> >
> > * I agree that the proposed method is effective for the Square attack, but the paper states that the method is proposed for black-box random search based attacks and there is a gap. The method is only evaluated for a single attack as I commented, and thus the paper does not show the advantage of the proposed method for general black-box random search based attacks.
> > Since it seems that the proposed method is suitable for the Square attack, It is still not clear whether the proposed method works for other attacks, especially for attacks which use simple random search with a few hyperparameters (e.g., only step size).
> >
> > * For the controller design, "task specific" means that the controllers are carefully designed for the Square attack. Although the architecture itself is simple, the input is not trivial and some techniques (e.g. relaxed sampling) are introduced. Since the method is proposed for various attacks, it is not desirable to need a lot of effort to design controllers. When people want to apply the proposed method to other attacks, they will need effort to design good controllers and need queries for the design.

---

### Official Review · Reviewer_mzdq · 2021-07-10

**Rating:** 6
**Confidence:** 4

**Summary:**

This paper proposes meta-learning a black box adversarial attack strategy. Specifically, the authors propose meta-learning controllers that output the parameters of a square attack. Experiments on CIFAR-10 and ImageNet demonstrate the effectiveness of a meta-learned square attack relative to a non meta-learned baseline square attack.

**Ethical Concerns:**

The authors may not have adequately described mitigation strategies for their proposed black-box adversarial attack (which may be a state-of-the-art attack). The proposed method could have several negative social impacts- for example attackers could use them to fool self-driving car systems and cause accidents. Because the proposed method is a black-box attack, the attackers do not even need internal access to the classifier.

**Ethics Review Area:**

["Inappropriate Potential Applications & Impact  (e.g., human rights concerns)"]

**Limitations And Societal Impact:**

The authors describe the limitations and social impact of this work in section 5. However, the authors do not adequately potential strategies that could be used to limit the negative social impact of adversarial attacks. This is particularly important if proposed method is a state-of-the-art black box adversarial attack. The authors may want to mention a few specific mitigation strategies to limit the negative impacts of their proposed attack.

**Main Review:**

**Originality**
As the authors point out, prior works have considered meta-learning attacks on neural networks. The authors propose meta-learning parameters of an existing black box attack. Nevertheless, the specific approach used by the authors appears novel.

**Quality**
The effectiveness of the proposed meta-learned square attack is evaluated experimentally on CIFAR-10 and ImageNet. However, there are some issues with the experiments:

First, the proposed method is framed as a meta-learned attack. However, the authors meta-train on a single dataset (CIFAR-10). Meta-learning typically refers to scenarios where meta-training occurs on a *set* of datasets and meta-testing occurs on a *different* set of datasets. The method proposed by the authors might be better framed as hyperparameter optimization or transfer learning rather than meta-learning. Alternatively, the authors might consider meta-training over a set of datasets and meta-testing on a different set of datasets.

Second, the improvements over baselines appear relatively incremental. In Table 1, it appears that meta-learning reduces the number of queries to achieve a certain accuracy by a factor of 2 or less. If the authors could reduce the number of queries by an order of magnitude of more, this would significantly improve the empirical experiments.

Relatedly, how does the proposed attack compared to other black box attacks (other than the square attack)? In particular, is the proposed method the state-of-the-art black box attack on CIFAR-10 or ImageNet (compared to previously proposed black box attacks)?

**Clarity**
The paper is well-written, with concepts introduced at appropriate points. Figures and tables are clear and appropriately sized. Experiments are adequately described.

**Significance**
Overall, the conceptual and empirical contributions in this paper appear relatively limited. Improvements over previous baselines seem to be small. If the authors can demonstrate that their results are a state-of-the-art black box attack, this would likely have practical value for neural network attackers and defenders. In general, the significance of this work depends strongly on the empirical improvement over prior works.

**Needs Ethics Review:**

Yes

**Time Spent Reviewing:**

2

---

> ### Author Response · Authors · 2021-08-10
> **Response to the raised concerns and questions**
>
> We thank the reviewer for the thorough review and helpful feedback. We address the open points below.
>
> Concerning the terminology, meta-learning typically refers to generalizing across a set of tasks [1], where these tasks can differ in terms of the data (as pointed out by the reviewer) but also in other aspects. In our case, the different tasks correspond to the different models whose robustness is evaluated. Thus, out work fits very well into the general setting of meta-learning, and more specifically "learning to optimize". However, we agree that meta-training over a set of datasets would be an interesting setting for our method and thank the reviewer for this suggestion. However, note that our learned controllers on CIFAR10 directly generalize to CIFAR100 (Table R2), ImageNet (Appendix A.1) or different attack modes such as targeted attacks (Table R3).
>
> We respectfully disagree that a reduction of the number of queries by a factor of roughly 2 (as observed by the reviewer) over a very competitive baseline such as square attack is "only" incremental.
>
> We considered the Square Attack because to the best of our knowledge it is the state-of-the-art method for the $\ell_\infty$ threat model in terms of query efficiency. Therefore, improving upon this method directly improves the state of the art. Comparison of Square Attack with the other recent attacks can be found e. g. in Table 5 in [2].
>
> Regarding the ethical concerns: we may add to the paper that  a general mitigation strategy could be to add a check to a system which rejects predictions for a sequence of inputs which all differ by at most an $\ell_{p}$ distance of some $\epsilon$. This would make most black-box attacks for image-specific perturbations ineffective, including meta-learned square attack.
>
> [1] Timothy Hospedales, Antreas Antoniou, Paul Micaelli, Amos Storkey, "Meta-Learning in Neural Networks: A Survey"
>
> [2] Yusuke Tashiro, Yang Song, Stefano Ermon. Diversity Can Be Transferred: Output Diversification for White- and Black-box Attacks, NeurIPS 2020

---

> > ### Comment · Reviewer_mzdq · 2021-08-29
> > **Thank you for your detailed response!**
> >
> > The authors convincingly argue that their method is a state-of-the-art black box attack in terms of query efficiency. Moreover, they address the ethical concerns associated with proposing a new state-of-the-art attack. Finally, their additional experiments are compelling. Thus, I am increasing my rating.

---

### Official Review · Reviewer_UUwL · 2021-07-11

**Rating:** 5
**Confidence:** 5

**Summary:**

The paper proposes a meta-learning approach to produce optimistic hyperparameters for score-based black-box adversarial attack, especially for Square Attack. A controller is trained on the source network to find the optimistic schedule of hyperparamters. The success rate is improved using the controller.

**Limitations And Societal Impact:**

See above.

**Main Review:**

**Pros**

1. Meta learning is a reasonable and promising way to improve the query efficiency and success rate of black-box attack. The attacker can use the information from the query and adjust itself to gain better performance.
2. The experiments, despite quite inadequate, do show that the hyperparameters of the Square Attack is sensitive to some hyperparameters and tuned hyperparameters can improve the query efficiency.

**Cons**

1. The paper fails to prove that the controller is able to generalize to unseen models. The models used in Table 1 are all adversarially trained model with similar loss landscapes. And the controller is trained on an adversarially trained model (Resnet18), which is similar to the models used for evaluation. I think the author should show that the controller is able to generalize to different settings such as attack for undefended models and targeted attack.
2. The author uses the information of the source model. Although the information is quite limited, the method should be compared with the methods combined transfer-based attack and score-based attack such as [1][2].
3. The author uses two inputs for the MLP. I think there should be an ablation study to show the how each input affect the finall success rate and query efficiency.
4. While the controller is promising direction for black-box attack, the authors only show its effectiveness on Square Attack. I would be better to show the controller is able to control other hyperparameters in different black-box attacks.


**Minors**

The notation $\mathcal{D}_\omega$ should be changed. It is easy to be mixed up with the notation $D$.

[1] Zhichao Huang, Tong Zhang. Black-box adversarial attack with transferable model-based embedding, ICLR 2020

[2] Yusuke Tashiro, Yang Song, Stefano Ermon. Diversity Can Be Transferred: Output Diversification for White- and Black-box Attacks, NeurIPS 2020

**Time Spent Reviewing:**

3

---

> ### Author Response · Authors · 2021-08-10
> **Clarification on Cons and Questions Raised**
>
> We would like to thank the reviewer for the helpful review and the constructive feedback. We will try to clarify the points mentioned in the Section "Cons":
>
> 1. "Generalization to unseen model and the diversity of the models architectures"
>
> Our goal was to test our attack on the SOTA of adversarially trained models for which reference results exist and thus we use models from robustbench. Almost all of them belong to a ResNet type.
>
> Among the models used for the evaluation in the Table 1 only one of has a ResNet18 architecture (same as the source model) but all other are relatively diverse: ResNet-50, WideResNet-28-4, WideResNet-28-10 and WideResNet-34-10 as stated in the Table 3 in the Appendix. We would also like to note that the robustness of each target model was obtained by its authors using a different training method or augmentation technique discussed in the corresponding papers. Therefore, we expect those models to have relatively different intrinsic aspects such as loss landscape.
>
> 2. "Relation to transfer-based and score-based attacks"
>
> Regarding [2] we would like to highlight that for untargeted attacks and l_inf-threat model they could not improve over the original Square Attack (Table 5 in [2]) - and we improve further Square Attack in this paper. We didn't manage to find the code of SimBA-ODS or ODS-RGF proposed in [2] to perform a more detailed comparison.
>
> We highlight that [1] uses a surrogate model for a transfer attack while we just use it for training the controller. Our learned controllers on CIFAR10 generalize directly to ImageNet (see Appendix A.1) whereas this is not feasible for [1].
>
> 3. "Effect of each input of the MLP and ablation study"
>
> The Figure 2 qualitatively illustrates the dependence of the controllers on the time (right pane) and the success rate (left pane) input.
>
> 4. "Effectiveness only shown on Square Attack"
>
> We have concentrated on the Square Attack because to the best of our knowledge it is the state-of-the-art method for the $\ell_\infty$ threat model in terms of query efficiency and success rate which is also supported by the recent work [2]. Besides that, it allows us to show an example of how the gradient-based end-to-end training of the controllers can be achieved when the attack has some discrete elements such as sampling squares of integer size. For additional experimental results concerning the Square Attack for $\ell_2$ threat model please refer to the Table R2.

---

> > ### Comment · Reviewer_UUwL · 2021-08-18
> > **Thanks for the response**
> >
> > **"Generalization to unseen model and the diversity of the models architectures"**
> >
> > I disagree with the authors on this point. The adversarially-trained models accuracy have similar loss landscape when comparing its difference of normally-trained models. I think the authors should try whether directly using the schedule of one model can increase the performance of other models.

---

> > > ### Author Response · Authors · 2021-08-20
> > > **Results for the normally-trained models with different architectures**
> > >
> > > We thank the reviewer for the response. We hope that the results provided below will clarify the raised issue.
> > >
> > > The learned controllers used in MSA are the ones trained on CIFAR10 that we use in our other experiments. Therefore, we demonstrate the generalization of these controllers to undefended models of different architectures in targeted and untargeted scenarios.
> > >
> > > Table R4. The results of the untargeted attacks on the undefended ImageNet models. We compare our method with the Square Attack [1] with parameter p=0.05 as suggested in the paper for this setting. We provide the robust accuracy estimate obtained for each of the considered query budgets.
> > >
> > > |Model                | Attack     | 500 q    | 1000 q   | 2500 q   | 5000 q      |
> > > |:--------------------|:-----------|:--------|:--------|:--------|:-----------|
> > > | ResNet-50           | SA         | 4.7     | 1.4     | **0.1**     | **0.0**        |
> > > |                     | MSA (our)  | **2.3**     | **0.7**     | **0.1**     | **0.0**        |
> > > |
> > > | VGG-16-BN           | SA         | 2.5     | 1.5     | **0.0**     | **0.0**        |
> > > |                     | MSA (our)  | **1.4**     | **0.2**     | **0.0**     | **0.0**        |
> > > |
> > > | Inception v3        | SA         | 9.3     | 3.5     | 1.2     | 0.2        |
> > > |                     | MSA (our)  | **3.7**     | **1.5**     | **0.5**     | **0.1**        |
> > >
> > > Table R5. The results of the targeted attacks on the undefended ImageNet models. We randomly choose the target class for each image. We compare our method with the Square Attack [1] with parameter p=0.01 as suggested in the paper for the targeted setting. We provide the robust accuracy estimate obtained for each of the considered query budgets. It is the fraction of the total images that was initially correctly classified by the model and not shifted to the target class during the attack.
> > >
> > > |Model                | Attack     | 500 q     | 1000 q   | 2500 q   | 5000 q      |
> > > |:--------------------|:-----------|:--------|:--------|:--------|:-----------|
> > > | ResNet-50           | SA         | 54.9    | 54.4    | 50.7    | 36.7       |
> > > |                     | MSA (our)  | **51.0**    | **46.0**    | **30.8**    | **15.7**       |
> > > |
> > > | VGG-16-BN           | SA         | 52.7    | 51.9    | 44.8    | 24.6       |
> > > |                     | MSA (our)  | **48.0**    | **39.9**    | **20.0**    | **7.5**        |
> > > |
> > > | Inception v3        | SA         | 57.3    | 56.9    | 54.1    | 46.6       |
> > > |                     | MSA (our)  | **54.3**    | **50.5**    | **41.9**    | **35.6**       |
> > >
> > > [1] Maksym Andriushchenko, Francesco Croce, Nicolas Flammarion, and Matthias Hein. Square attack: a query-efficient black-box adversarial attack via random search. In ECCV, 2020.

---

> > > > ### Comment · Reviewer_UUwL · 2021-08-23
> > > > **Follow-up comments**
> > > >
> > > > Thanks for the detailed experiments. I will increase my score to 5.

---

### Official Review · Reviewer_KmCh · 2021-07-14

**Rating:** 6
**Confidence:** 4

**Summary:**

The authors propose a meta-learning framework to improve the query efficiency for Square Attack.


**Limitations And Societal Impact:**

Yes

**Main Review:**

This paper focus on improving the query efficiency of a score-based black-box attack named Square Attack.
Specifically, the authors make some hyper-parameters of Square Attack (i.e., square size and the color channel) to be adaptive.
The authors use current step size, the history of loss values and the success frequencies of different color channels as inputs, and design MLPs to map from these inputs to better values of square size and color channel.
The MLPs are then trained using a meta-learning formulation, to make the learned strategies (parameterized by MLPs) to have better generalization performance on unseen images and models.

Strengths:
* The proposed method seems to be technically sound and novel. The square size and color channel schedules in Square Attack are crucial for query efficiency, but these parameters require significant manual design. The authors propose to directly learn these parameters via meta-learning, such that the learned schedules can work well on unseen images and models.
* The controllers are trained on a white-box ResNet-18 model, and are evaluated on black-box RobustBench models. As such, an attacker does not need to know the detailed architecture of the target model, making the proposed method more practical.

Weakness:
* The authors claim in the abstract that the controller "can be plugged into random search methods", however, the proposed method seems to be deeply coupled with one particular attack (i.e., Square Attack). Can the proposed method be easily integrated with other attacks?
* In line 38-39, the authors said "...propose a method to ...reduce the amount of manual design in random search based attacks.". However, the proposed two controllers themselves also require significant manual design (in Section 3.3).
* In Table 1, although the proposed method has better query efficiency than competitive methods, the differences are not great in my opinion, given the fact that competitive methods use manually designed schedules while the proposed method use learned schedules. I wonder whether the advantage of the proposed method is more clear in extremely low query budget regime (e.g., < 150 queries), since in the early stage, the accuracies have not been saturated, thus a good search distribution could possibly make more contribution to the query efficiency.


**Time Spent Reviewing:**

25

---

> ### Author Response · Authors · 2021-08-10
> **Addressing the Section "Weaknesses"**
>
> We would like to thank the reviewer for the thorough review and the provided suggestions. We will try to clarify the points mentioned in the Section "Weaknesses":
>
> - "The method is deeply coupled with the Square Attack"
>
> The inputs used by the controllers such as the update success frequency and the current search iteration are common for a broad family of random-search based methods. Therefore,  the general methodology outlined in the Section 3.2 is applicable to other random search-based attacks. However, the requirement of end-to-end differentiation might require attack-specific relaxation techniques.
>
> - "Significant manual design of the controllers"
>
> As discussed in the Section 4.1, the controller itself, which is an MLP with 2 layers and 10 hidden units each, is quite simple and was purposefully not fine-tuned. The design of the controllers' inputs discussed in Section 3.3 (that might seem complex) allows the inputs to be in the [0, 1] range which enhances the end-to-end training. The ability of this input form to generalize to the other settings is supported by the ImageNet transfer results in the Table 2 in the Appendix as well as the results provided in the Tables R1, R2, R3.
>
> - "Comparison in extremely low regime"
>
> The following table provides the results for the query budget limited by 100 queries (all the experimental details are identical to the ones described in the Table 1 in the paper).
>
> |Square size  | Color      | Wong et al.      |  Ding et al.      | Engstrom et al.   | Gowal et al.      | Carmon et al.     | Huang et al.     |
> |:------------|:-----------|-----------------:|------------------:|------------------:|------------------:|------------------:|-----------------:|
> | SA          | Uni        |78.8$\pm$0.15     | 78.8$\pm$0.14     | 82.9$\pm$0.13     | 87.1$\pm$0.16     | 86.5$\pm$0.09     | 80.9$\pm$0.07    |
> | AA          | Uni        |74.4$\pm$0.07     | 73.3$\pm$0.17     | 78.0$\pm$0.20     | 83.7$\pm$0.16     | 83.2$\pm$0.10     | 76.3$\pm$0.14    |
> | $MSA_s$     | Uni        |**73.0$\pm$0.07** | **71.4$\pm$0.10** | **75.9$\pm$0.06** | **82.5$\pm$0.07** | **81.4$\pm$0.07** | **74.7$\pm$0.10**|
> | $MSA_s$     | $MSA_c$    |73.2$\pm$0.12     | 71.5$\pm$0.12     | 76.4$\pm$0.18     | 82.6$\pm$0.06     | 81.6$\pm$0.07     | 75.0$\pm$0.07    |

---

### Review · Ethics_Reviewer_8PRu · 2021-07-23

**Recommendation:**

The paper is in reasonable shape from an ethical review perspective, considering the paper acknolwedges the potential harms due to adversairal attacks but points out that long term benefits outweight medium term risks.

**Ethics Review:**

The paper includes improved methods for adversarial attacks. Though adversarial attacks have potentail for downstream harm, the authors include reasonable discussion surrounding the utility of developing adversarial attacks in their impact section. Specifically, they point out that  long term benefits outweigh short/medium term downsides of revealing vulnerabilities. This is a reasonable point and is generally accepted in this field. Thus, the paper does not raise ethical concerns.

---

### Review · Ethics_Reviewer_wrLg · 2021-08-09

**Recommendation:**

These issues could be addressed with a more substantial engagement with the mitigation strategies that would reduce potential negative impacts. If there are none at present, that should be noted as well - it's an important part of weighing the benefits and risks of this approach.

**Ethical Issues:**

Yes

**Ethics Review:**

The paper proposes an approach to improve query efficiency of black-box random search based adversarial attacks, specifically square attack. I agree with the technical reviewer that observed that the authors have not adequately described mitigation strategies for their proposed black-box adversarial attack - a serious concern if this approach were to be used in high-stakes contexts like self-driving cars, robotics, drones, etc, and also if it were used to generate noise to bypass content filters. This depends on whether the paper is sufficiently novel to constitute state-of-the-art, which is beyond the scope of the ethical review.

---

### Author Response · Authors · 2021-08-10
**Experimental results showing the generalization of the method and the learned controllers**

We would like to thank all the reviewers for the helpful reviews and the constructive feedback. To address the recurring points concerning the similarity between the source and the target models and testing the method for different datasets and attacks, we would like to demonstrate the following experimental results:

Table R1. Results of the $\ell_\infty$ Square Attack with $\epsilon=8/255$ on CIFAR100 models obtained by applying the controllers meta-trained for the CIFAR10 model thereby demonstrating their generalization to a different dataset (supporting the results for the ImageNet generalization provided in the Table 2 in the Appendix). The results are averaged across 3 runs with different random seeds.

|Model                | Clean acc. | Robust acc. | Square size | Color   | 500               | 1000              | 2500              | 5000              |
|:--------------------|:-----------|:------------|:------------|:--------|:------------------|:------------------|:------------------|:------------------|
| Wu et al. [1]       | 60.38      | 28.86       | SA          | Uni     | 43.3$\pm$0.17     | 38.7$\pm$0.09     | 33.9$\pm$0.28     | 32.3$\pm$0.20     |
|                     |            |             | AA          | Uni     | 41.3$\pm$0.20     | 37.6$\pm$0.00     | 35.0$\pm$0.06     | 32.7$\pm$0.35     |
|                     |            |             | $MSA_s$     | Uni     | 38.0$\pm$0.06     | 35.8$\pm$0.18     | 33.5$\pm$0.03     | 32.4$\pm$0.06     |
|                     |            |             | $MSA_s$     | $MSA_c$ | **37.8$\pm$0.07** | **35.5$\pm$0.12** | **33.1$\pm$0.03** | **32.2$\pm$0.03** |
|
| Cui et al. [2]      | 70.25      | 27.16       | SA          | Uni     | 48.9$\pm$0.03     | 42.3$\pm$0.20     | 33.6$\pm$0.09     | 30.5$\pm$0.06     |
|                     |            |             | AA          | Uni     | 47.5$\pm$0.17     | 42.8$\pm$0.09     | 35.9$\pm$0.13     | 32.5$\pm$0.15     |
|                     |            |             | $MSA_s$     | Uni     | 43.2$\pm$0.24     | 38.6$\pm$0.24     | 32.8$\pm$0.21     | 31.0$\pm$0.09     |
|                     |            |             | $MSA_s$     | $MSA_c$ | **42.6$\pm$0.13** | **37.8$\pm$0.12** | **32.5$\pm$0.30** | **30.1$\pm$0.09** |

Table R2. Results for the $\ell_2$ Square Attack with $\epsilon=0.5$. To show the generalization of our method to the random-search based attacks other than the $\ell_\infty$ Square Attack, we have considered the Square Attack for the $\ell_2$ case. Although the update procedure of these two methods is structurally different (cf. Algorithm 1 and Algorithm 3 in [3]), we managed to achieve the end-to-end differentiation by using a similar update size interpolation scheme as in the Section A.3. We have tested the obtained update size controller $MSA_s^2$ on the robust $\ell_2$ models from robustbench. The results are averaged across 3 runs with different random seeds. Note that although the schedule denoted as SA was dominating the schedule denoted as AA for most of the query budgets in the $\ell_\infty$ case (see Table 1 in the paper), for the $\ell_2$ we have the opposite situation, which demonstrates the sensitivity of the different random-search based attacks to the underlying parameters (even though they both belong to a general "Square Attack" family).

|Model                | Clean acc. | Robust acc. | Attack      | Color   | 500               | 1000              | 2500              | 5000              |
|:--------------------|:-----------|:------------|:------------|:--------|:------------------|:------------------|:------------------|:------------------|
| Ding et al. [4]     | 88.02      | 66.09       | SA          | Uni     | 85.6     | 83.7     | 81.2     | 78.2     |
|                     |            |             | AA          | Uni     | 82.8$\pm$0.09     | 81.4$\pm$0.06     | 79.2$\pm$0.00     | 77.7$\pm$0.15     |
|                     |            |             | $MSA_s^2$   | $MSA_c$ | **82.3$\pm$0.03** | **80.9$\pm$0.07** | **77.4$\pm$0.09** | **75.8$\pm$0.19** |
|
| Rice et al. [5]     | 88.67      | 67.68       | SA          | Uni     | 86.1     | 84.8     | 81.3     | 79.7     |
|                     |            |             | AA          | Uni     | 83.7$\pm$0.12     | 81.4$\pm$0.09     | 79.9$\pm$0.09     | 78.6$\pm$0.18     |
|                     |            |             | $MSA_s^2$   | $MSA_c$ | **82.6$\pm$0.09** | **81.0$\pm$0.07** | **78.7$\pm$0.03** | **76.9$\pm$0.25** |
|
| Augustin et al. [6] | 91.08      | 72.91       | SA          | Uni     |  89.0    | 88.4     | 86.9     | 84.2     |
|                     |            |             | AA          | Uni     | 87.8$\pm$0.03     | 86.8$\pm$0.09     | 84.8$\pm$0.17     | 83.3$\pm$0.17     |
|                     |            |             | $MSA_s^2$   | $MSA_c$ | **87.5$\pm$0.12** | **86.3$\pm$0.06** | **83.4$\pm$0.03** | **81.8$\pm$0.07** |
|
| Engstrom et al. [7] | 90.83      | 69.24       | SA          | Uni     | 87.3     | 86.1     | 84.0      | 80.8      |
|                     |            |             | AA          | Uni     | 85.3$\pm$0.06     | 83.7$\pm$0.15     | 81.5$\pm$0.24     | 78.6$\pm$0.06     |
|                     |            |             | $MSA_s^2$   | $MSA_c$ | **84.7$\pm$0.09** | **83.1$\pm$0.06** | **79.7$\pm$0.13** | **77.4$\pm$0.00** |
|
| Rony et al. [8]     | 89.05      | 66.44       | SA          | Uni     | 85.4     | 83.5     | 80.5      | 78.3      |
|                     |            |             | AA          | Uni     | 82.0$\pm$0.10     | 80.8$\pm$0.10     | 78.9$\pm$0.03     | 77.0$\pm$0.15     |
|                     |            |             | $MSA_s^2$   | $MSA_c$ | **81.6$\pm$0.03** | **80.4$\pm$0.00** | **77.2$\pm$0.00** | **75.7$\pm$0.09** |

Table R3. Results for the $\ell_\infty$ targeted attack with $\epsilon=8/255$ on CIFAR10 with a query budget limit of 10000 queries. The controllers used in the evaluation were trained in the untargeted setting thereby we demonstrate their generalization to the targeted case. In this scenario of a big query budget Square Attack schedule $(p=0.3)$ does not require rescaling.

|Model                | Attack              | Accuracy | Average queries | Median queries |
|:--------------------|:--------------------|---------:|----------------:|---------------:|
| Wong et al.         | Square ($p^0=0.8$)  | 75.5     | 2978            | 1302           |
|                     | Square ($p^0=0.3$)  | **75.2** | 2247            | 951            |
|                     | $MSA_s$             | 75.6     | 2005            | 793            |
|                     | $MSA_s + MSA_c$     | 75.3     | **1948**        | **771**        |
|
| Ding et al.         | Square ($p^0=0.8$)  | 73.1     | 2359            | 1329.5         |
|                     | Square ($p^0=0.3$)  | **72.9** | 1666            | 773.5          |
|                     | $MSA_s$             | 73.5     | 1551            | 722            |
|                     | $MSA_s + MSA_c$     | 73.1     | **1518**        | **646.5**      |
|
| Engstrom et al.     | Square ($p^0=0.8$)  | 77.8     | 2295            | 1185.5         |
|                     | Square ($p^0=0.3$)  | 77.2     | 1818            | 810            |
|                     | $MSA_s$             | 77.4     | 1988            | **702.5**      |
|                     | $MSA_s + MSA_c$     | **77.1** | **1810**        | 757            |
|
| Gowal et al.        | Square ($p^0=0.8$)  | 85.0     | 2196            | 1069           |
|                     | Square ($p^0=0.3$)  | **84.8** | 1738            | 665            |
|                     | $MSA_s$             | 84.9     | 1625            | **519**        |
|                     | $MSA_s + MSA_c$     | **84.8** | **1466**        | 540.5          |
|
| Carmon et al.       | Square ($p^0=0.8$)  | 83.9     | 2272            | 1246.5         |
|                     | Square ($p^0=0.3$)  | **83.7** | 1572            | 776            |
|                     | $MSA_s$             | 84.0     | **1449**        | 694            |
|                     | $MSA_s + MSA_c$     | **83.7** | 1541            | **657**        |
|
| Huang et al.        | Square ($p^0=0.8$)  | 77.8     | 1652            | 795.5          |
|                     | Square ($p^0=0.3$)  | **77.6** | 1242            | 531            |
|                     | $MSA_s$             | 77.7     | 1126            | **353**        |
|                     | $MSA_s + MSA_c$     | **77.6** | **1076**        | 414            |


[1] Dongxian Wu, Shu tao Xia, and Yisen Wang. Adversarial weight perturbation helps robust generalization. In NeurIPS, 2020.

[2] Jiequan Cui, Shu Liu, Liwei Wang, and Jiaya Jia. Learnable boundary guided adversarial training, 2020.

[3] Maksym Andriushchenko, Francesco Croce, Nicolas Flammarion, and Matthias Hein. Square attack: a query-efficient black-box adversarial attack via random search. In ECCV, 2020.

[4] Gavin Weiguang Ding, Yash Sharma, Kry Yik Chau Lui, and Ruitong Huang. MMA training: Direct input space margin maximization through adversarial training. In ICLR, 2020.

[5] Leslie Rice, Eric Wong, and J. Zico Kolter. Overfitting in adversarially robust deep learning. In ICML, 2020.

[6] Maximilian Augustin, Alexander Meinke, and Matthias Hein. Adversarial robustness on in and out-distribution improves explainability. In ECCV, 2020.

[7] Logan Engstrom, Andrew Ilyas, Hadi Salman, Shibani Santurkar, and Dimitris Tsipras. Robustness (python library), 2019.

[8] Jérôme Rony, Luiz G. Hafemann, Luiz S. Oliveira, Ismail Ben Ayed, Robert Sabourin, and Eric Granger. Decoupling direction and norm for efficient gradient-based l2 adversarial attacks and defenses. In CVPR, 2019.

---

### Public Comment · ~Maksym_Yatsura1 · 2021-11-16
**Fixing an inaccuracy in Table 5 in the paper**

We want to notify the readers that due to an experimental pipeline flaw that we have recently discovered the results provided in Table 5 in the paper were obtained by running the torchvision models on unnormalized data i. e. the input pixels were in the range [0, 1]. Therefore the clean accuracy of the models was lower than it was supposed to be.

We provide the corresponding results with normalized data used. The big improvement of MSA (more than 20 percent points) with respect to SA with p=0.01 for the targeted setting still holds.

Table 5. MSA trained on a CIFAR10 model attacking the undefended ImageNet models in the $\ell_\infty$ threat model with $\epsilon=0.05$. We compare our method with SA and set $p^0=0.05$ for the untargeted case and $p^0=0.01$ for the targeted case as suggested by Andriushchenko et al. [1]. For the targeted attacks robust accuracy is the fraction of the total number of images that was initially correctly classified by the model and not shifted to the target class during the attack. We provide clean accuracy of the models on a subset of 1000 ImageNet validation set images that we consider.

Untargeted

| Model		      | Clean acc. (%) | Attack	| 500       | 1000    | 2500    | 5000    |
|:--------------------|:---------------|:-------|:----------|:--------|:--------|:--------|
| ResNet-50 [2]       | 77.3           | SA     | 8.8       | 5.1     | 0.2     | **0.0** |
|                     |                | MSA    | **2.9**   | **0.8** | **0.0** | **0.0** |
|
| VGG16-BN [3]        | 75.0           | SA     | 2.8       | 0.9     | 0.0     | **0.0** |
|                     |                | MSA    | **1.8**   | **0.2** | **0.0** | **0.0** |
|
| Inception v3 [4]    | 77.6           | SA     | 16.6      | 6.1     | **2.3** | **1.0** |
|                     |                | MSA    | **10.2**  | **5.1**  | 2.6     | 1.3     |

Targeted

| Model		      | Clean acc. (%) | Attack	| 500       | 1000    | 2500    | 5000    |
|:--------------------|:---------------|:-------|:----------|:--------|:--------|:--------|
| ResNet-50 [2]       | 77.3           | SA     | 76.9      | 75.1    | 62.5    | 34.4    |
|                     |                | MSA    | **67.1**  | **52.0**| **27.8**| **12.1**|
|
| VGG16-BN [3]        | 75.0           | SA     | 74.5      | 72.2    | 51.5    | 17.4    |
|                     |                | MSA    | **62.5**  | **45.2**| **16.5**| **3.5** |
|
| Inception v3 [4]    | 77.6           | SA     | 77.5      | 76.4    | 70.9    | 59.8    |
|                     |                | MSA    | **74.4**  | **70.6**| **60.1**| **49.5**|

[1] Maksym Andriushchenko, Francesco Croce, Nicolas Flammarion, and Matthias Hein. Square attack: a query-efficient black-box adversarial attack via random search. In ECCV, 2020.

[2] Kaiming He, Xiangyu Zhang, Shaoqing Ren, and Jian Sun. Deep residual learning for image recognition. In CVPR, 2016.

[3] Karen Simonyan and Andrew Zisserman. Very deep convolutional networks for large-scale image recognition. In ICLR, 2015.

[4] Christian Szegedy, Vincent Vanhoucke, Sergey Ioffe, Jonathon Shlens, and Zbigniew Wojna. Rethinking the inception architecture for computer vision. In CVPR, 2016.

---

### Decision · Program_Chairs · 2021-09-27

**Decision:**

Accept (Poster)

**Comment:**

First, we would like to commend the authors on an interesting submission and a heroic effort in the response period, including several experimental findings to help address reviewer concerns. It is extremely difficult to make a decision for this paper --- it is on the borderline. The scores are all borderline, tending on average towards reject, without a clear champion. However, the authors do make a reasonably good case that they are proposing a query efficient black box attack. In the rebuttal they also attempt to address ethical concerns. On the other hand, there are remaining concerns about the technical contribution, particularly how specific the approach is to the square attack. The questions are mostly around how applicable the ideas are to other attacks, and how much effort would be required to design controllers for other attacks. In any case, the effort the reviewers have made in responding to reviewers should be highly useful in subsequent revisions.